# CAN LARGE LANGUAGE MODELS REASON?
# A CHARACTERIZATION VIA 3-SAT

## ABSTRACT

Large Language Models (LLMs) have been touted as AI models possessing advanced reasoning abilities. However, recent works have shown that LLMs often bypass true reasoning using shortcuts, sparking skepticism. To study the reasoning capabilities in a principled fashion, we adopt a computational theory perspective and propose an experimental protocol centered on 3-SAT – the prototypical NP-complete problem lying at the core of logical reasoning and constraint satisfaction tasks. Specifically, we examine the phase transitions in random 3-SAT and characterize the reasoning abilities of LLMs by varying the inherent hardness of the problem instances. Our experimental evidence shows that LLMs are incapable of performing true reasoning, as required for solving 3-SAT problems. Moreover, we observe significant performance variation based on the inherent hardness of the problems – performing poorly on harder instances and vice versa. Importantly, we show that integrating external reasoners can considerably enhance LLM performance. By following a principled experimental protocol, our study draws concrete conclusions and moves beyond the anecdotal evidence often found in LLM reasoning research.

## 1 INTRODUCTION

The success and versatility of Large Language Models (LLMs) have sparked widespread interest and debate on whether LLMs are capable of reasoning. The answer to this question may depend on the perspective on reasoning one takes, whether it is more oriented toward common sense reasoning (Davis & Marcus, 2015) or towards logical or deductive reasoning (Genesereth & Nilsson, 1987). We will adhere to Leon Bottou's definition, which defines reasoning as "*algebraically manipulating previously acquired knowledge in order to answer a new question*" (Bottou, 2014). This is aligned with Russell and Norvig's description of artificial intelligence as rational thinking (Russell & Norvig, 2010).

Recent studies suggest that LLMs are inherently capable of zero-shot reasoning (Kojima et al., 2022) (i.e. performing multistep inference processes in previously unseen situations). This ability has been shown to *emerge* and improve with scale (Wei et al., 2022a; Srivastava et al., 2023), and can be further enhanced by using smart prompting techniques that encourage LLMs to think step-by-step (Kojima et al., 2022; Wei et al., 2022b; Zhou et al., 2023; Yao et al., 2023b; Prasad et al., 2023). Demonstrations include, inter alia, planning (Huang et al., 2022; Ahn et al., 2022; Singh et al., 2023; Liang et al., 2023; Huang et al., 2023; Hazra et al., 2024b), theorem proving (Jiang et al., 2023; Welleck et al., 2022; Yang et al., 2022), search and optimization (Yang et al., 2024; Romera-Paredes et al., 2024; Hazra et al., 2024a), self-reflection (Yao et al., 2023a; Madaan et al., 2023; Shinn et al., 2023), and tool usage (Shen et al., 2023; Schick et al., 2023).

Conversely, a growing body of research presents a more critical view of these emergent abilities. For instance, LLMs may exhibit limitations in consistent logical reasoning (Arkoudas, 2023; Saparov & He, 2023), effective planning (Valmeekam et al., 2022), and accurate self-evaluation of their outputs (Stechly et al., 2023). During training, language models can fit on statistical features (Zhang et al., 2023) or identify reasoning shortcuts, much like the *Clever Hans Cheat* (Bachmann & Nagarajan, 2024), thus bypassing true reasoning. There is also growing concern about dataset contamination[1] from open-source benchmarks (Zhang et al., 2024) that can inflate the reasoning performance

---

[1]Data closely resembling the benchmark leaks into the training data.

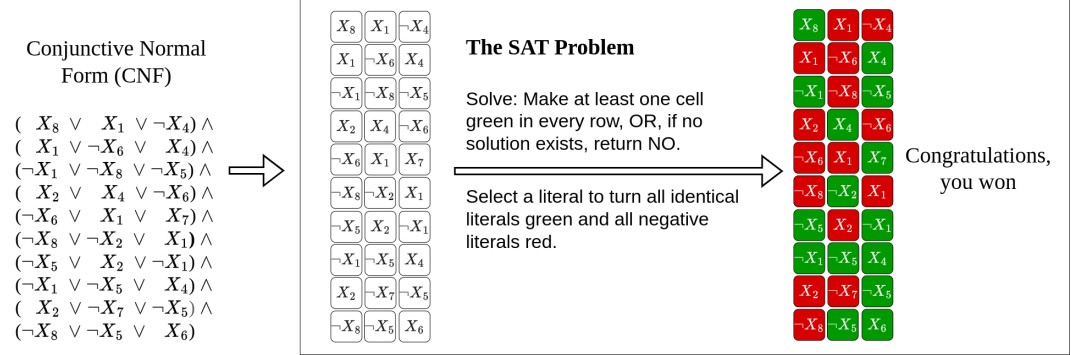

Figure 1: **The 3-SAT problem**, visualized using a variant of the SAT game (Roussel). In SAT, one must return a truth assignment to Boolean variables that satisfy a Boolean formula in conjunctive normal form (CNF) if one exists, and return unSAT otherwise. A row in the visualization represents a clause, which is a disjunction (connected by a logical OR $\vee$) of literals, wherein a literal can be positive ($X_1$) or negative ($\neg X_1$). An assignment satisfies a clause if one of its literals is assigned the value true. Since clauses are conjunctively connected (by a logical AND $\wedge$), all clauses must be satisfied for the formula to be satisfied. If no satisfying assignment exists, the formula is unsatisfiable.

of LLMs. During inference, the autoregressive nature of LLMs makes them prone to snowballing errors over time (Dziri et al., 2023). Furthermore, these models can often generate unfaithful and biased explanations in their chains of thought (Turpin et al., 2023). Additionally, their greedy approach to reasoning often falls short in contexts with multiple valid reasoning steps (Saparov & He, 2023). On the architectural side of LLMs, findings reveal that the transformer layer is incapable of function composition for large domains (Peng et al., 2024; Dziri et al., 2023). From a theoretical standpoint, transformer computations have been shown to lie in the complexity class log-uniform $TC^0$, which is too restrictive even for simple logical reasoning tasks such as 2-SAT[2]. Given these limitations, it has been suggested that the emergent abilities are but a mere *mirage* stemming from the choice of metric (Schaeffer et al., 2023). This inevitably leads to the question: "**Can Large Language Models reason?**" And if so, to what extent? We answer these questions via the following contributions.

**(1) Characterizing LLM-reasoning from a computational theory perspective.** Adhering to Leon Bottou's definition of reasoning, we propose an experimental framework centered on 3-SAT, to evaluate reasoning. The choice of 3-SAT, introduced in Figure 1, is not arbitrary – being a foundational problem in computational complexity, many problems in AI such as (propositional fragments of) logical reasoning, planning, and constraint satisfaction can be reduced to 3-SAT. Current benchmarks (Hendrycks et al., 2021a; Chen et al., 2021; Hendrycks et al., 2021b; Rein et al., 2023) often conflate commonsense reasoning (which involves knowledge retrieval), with logic and deductive reasoning (which requires algebraic manipulation of knowledge as per Bottou's definition). This conflation makes it challenging to isolate the logical reasoning abilities of LLMs. Instead, our approach provides a more formal and robust evaluation of reasoning abilities. Additionally, it bridges classical computational theory with modern AI, allowing us to assess if LLMs go beyond pattern recognition and exhibit true reasoning capabilities grounded in complexity theory.

**(2) Studying Phase Transition Characteristics of LLMs.** We investigate the phase transition characteristics of LLMs (Cheeseman et al., 1991) – i.e., how LLMs' performance varies in the easy and hard regions of the problem space. This contrasts with commonly used reasoning benchmarks, which often evaluate LLMs without considering the performance variations based on *inherent hardness* of the problems.

---

[2]More specifically, $TC^0$ is more restrictive than the logarithmic memory class L (Papadimitriou, 1994, Chapter 16), (Merrill & Sabharwal, 2023). Given that the 2-SAT problem is NL-complete (i.e., the class of non-deterministic logarithmic space algorithms), multi-layer transformers cannot solve 2-SAT instances, unless L=NL (Peng et al., 2024).

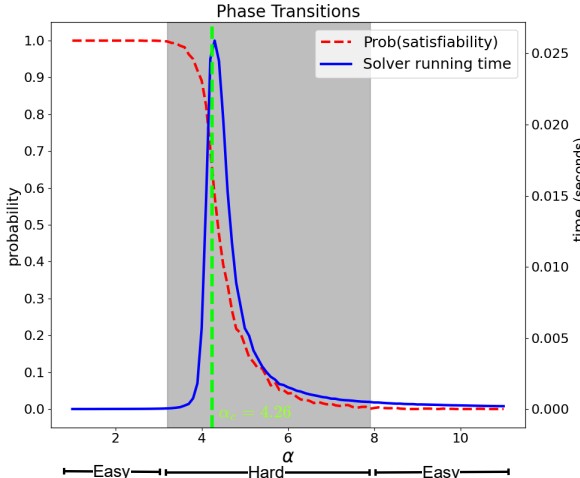

Figure 2: **Random 3-SAT Phase Transitions** (Cheeseman et al., 1991). Plotted in red is the probability of a randomly sampled 3-SAT formula being satisfied against the hardness $\alpha$ of the formula. We can observe a clear phase transition occurring at $\alpha_c \approx 4.267$ (marked by the green dotted line). We identify two easy regions, one on either side of $\alpha_c$. The gray area in the middle denotes the hard region. The boundaries of the hard region are defined where the probability of the formula being satisfied ceases to be deterministically one (left) or zero (right). The solid blue line shows the mean time taken by the MiniSAT solver (Eén & Sörensson, 2003) to solve a 3-SAT instance. Notably, there is a spike in the solver's runtime near the critical $\alpha_c$ value. This is attributed to the absence of useful heuristics in this region, forcing the solver to resort to essentially exhaustive searches.

**(3) Comprehensive evaluation of state-of-the-art open-source and proprietary LLMs.** We conducted extensive experiments across a range of state-of-the-art open-source LLMs, as well as proprietary LLMs. While we find that LLMs generally cannot reason, their performance varies significantly in different regions of problem space (i.e. phase transition characteristics), with GPT-4 significantly outperforming other models. Additionally, we demonstrate how a straightforward integration of LLMs with external verifiers, as demonstrated in (Ye et al., 2024; Liu et al., 2023; Wong et al., 2023), can enhance reasoning capabilities and improve performance on 3-SAT problems.

## 2 Preliminaries

### 2.1 Phase Transitions in Random 3-SAT

We study the reasoning capabilities of LLMs on random 3-SAT problems. 3-SAT constitutes one of the most fundamental problems in computer science as it is the prototypical NP-complete problem, lying at the foundation of computational complexity theory. Moreover, various prevalent reasoning problems in artificial intelligence, such as planning and constraint satisfaction, can be reduced to solving 3-SAT problems (Garey & Johnson, 1990).

By randomly sampling 3-SAT formulas[3], we avoid domain-specific biases, and it lets us control the complexity of the generated formulas. In fact, an interesting empirical observation is the presence of a *phase transition* in random 3-SAT problems (Cheeseman et al., 1991). When randomly sampling 3-SAT formulas, one can observe a sharp change in the probability of a 3-SAT formula being satisfiable when plotted against $\alpha = m/n$, where $m$ is the number of clauses and $n$ is the number of variables. For random 3-SAT, this phase transition occurs at $\alpha_c \approx 4.267$ (Mertens et al., 2006; Ding et al., 2015), i.e. the point at which a randomly sampled 3-SAT formula has equal probability to be satisfiable or unsatisfiable. This naturally divides 3-SAT problems into three regions: the

---

[3]We use Selman et al. (1996) random model, where clauses are sampled with replacement.

under-constrained region below the threshold (Easy), the constrained region in the neighborhood of the threshold (Hard), and the over-constrained region above the threshold (Easy), cf. Figure 2.

## 2.2 SAT SOLVERS

SAT solvers are tools to automatically verify the satisfiability of propositional logic formulas. The Davis–Putnam–Logemann–Loveland (DPLL) algorithm (Davis et al., 1962) is a key component of modern SAT solvers. It consists of a backtracking-based search algorithm, enriched with deductive rules and heuristics, to efficiently explore the search space and determine whether a formula is satisfiable. The core backtracking algorithm simply selects a literal and assigns a truth value to it. This step produces a simplified formula where all the parts made true by the assignment are removed. This step can be applied recursively until all clauses are removed, meaning that the original formula is satisfiable. Modern SAT solvers are based on Conflict-Driven Clause Learning (CDCL) (Silva & Sakallah, 1996) which enhances DPLL with conflict analysis and clause learning. In Figure 2 we show how the time to determine the satisfiability of a random 3-SAT formula varies in function of $\alpha$. Most prominently, we see a pronounced peak in time-to-solution around the $\alpha_c$.

Analogously to characterizing SAT solvers by their behavior on solving problems with varying $\alpha$, we study the reasoning capabilities of LLMs with respect to the phase transition in random 3-SAT problems.[4]. This contrasts with other works that characterize reasoning by evaluating performance on benchmark datasets (Cobbe et al., 2021; Hendrycks et al., 2021a).

## 3 RELATED WORK

Existing works have focused on improving the reasoning abilities of LLMs by combining them with verifiers and heuristics (Olausson et al., 2023; Liu et al., 2023; Lightman et al., 2024). In terms of solving SAT problems with LLMs, SATLM (Ye et al., 2024) proposes a satisfiability-aided language modeling using an LLM to parse an NL SAT input to a SAT problem specification which consists of a set of logical formulas, then obtain the solution by invoking a SAT solver. Similarly, AutoSAT (Sun et al., 2024) uses LLMs to optimize heuristics in SAT solvers. Our goal is *not* to build 3-SAT solvers powered by LLMs. On the contrary, we use the 3-SAT phase transition to assess the reasoning abilities of LLMs using a well-established experimental protocol. Closest to our work is the NPHardEval (Fan et al., 2024), which examines reasoning across various computational complexity classes, but we additionally explore phase transition characteristics and investigate how the performance varies with *inherent hardness* of problems.

Our work also aligns with the findings of Dziri et al. (2023) that investigates the performance of LLMs on compositional tasks with varying levels of complexity. Their experiments reveal a significant performance decline as task complexity increases, measured by problem size and reasoning depth. The findings indicate that while models can memorize single-step operations, potentially due to their prevalence during training, they fail to compose these steps into correct reasoning paths effectively. Similar observations were made on BERT-based models for a computationally tractable class of problems (not NP-complete) (Zhang et al., 2023). Our work extends these findings by demonstrating that the observed performance decline is better explained by the inherent hardness of problems, as defined by phase transitions, rather than problem size or depth. Large 3-SAT formulas with many variables or clauses may belong to either easy or hard regions. However, in easy regions, LLMs can still exploit statistical patterns, while in hard regions, they struggle due to the absence of effective heuristics. Similarly, statistical features can help reduce search depth in easy regions, but in hard regions, LLMs are forced to perform multi-step reasoning, revealing their limitations.

Merrill & Sabharwal (2023) established that multi-layer transformers belong to the complexity class log-uniform $TC^0$. Subsequently, Peng et al. (2024) demonstrated that multi-layer transformers cannot solve problems such as Derivability, 2-SAT, Horn SAT, and Circuit Evaluation unless L=NL. While these results provide worst-case performance bounds for transformer architectures, their relevance to average-case complexity is limited (Coarfa et al., 2000), and therefore, they offer only partial insights into the practical reasoning capabilities of LLMs. Notably, recent works have shown that $T$ chain of thought (CoT) steps can extends transformers' abilities beyond $TC^0$, up to those solvable by Boolean circuits of size T (Li et al., 2024). Our experiments on 3-SAT complement these

---

[4]A thought experiment in the same vein was also proposed by Kambhampati et al. (2024b)

findings, showing that GPT-4 allocates more tokens to hard problem regions, suggesting apparent reasoning despite limited overall performance.

## 4 METHODOLOGY

### 4.1 USING LLMS AS 3-SAT SOLVERS

To use LLMs as 3-SAT solvers, we reframe the 3-SAT problem as a natural language menu-selection problem, termed as **SAT-Menu**. As shown in Box 1, the prompt input to the LLM consists of a task outline, along with a specific scenario detailing the dietary preferences of a set of people. The LLM's objective is to identify a combination of orderable (akin to positive literals) and non-orderable (akin to negative literals) food items that meet these preferences; or declare the situation unsatisfiable (unSAT) if no valid combination exists. Note, that the prompt example in Box 1 constitutes a minimal example stripped of all details. The complete system prompt incorporates techniques known to enhance the apparent reasoning capabilities of LLMs, such as chain-of-thought (CoT) (Wei et al., 2022b) and in-context learning (Brown et al., 2020) (see Box 2 in Appendix for the full prompt).

Additionally, we introduce a second problem formulation where the LLM is directly given the underlying 3-SAT formula in Conjunctive Normal Form (CNF). We refer to this scenario as **SAT-CNF**. Specifically, in this setting, the problem is presented as a list of integers to the LLM, similar to the approach outlined in SAT Game (Figure 1). For more details about the prompt, we refer the reader to Box 3 in the Appendix.

---

**Box 1: SAT-Menu Prompt**

`# System Message`
Your task is to output two distinct lists of food items, one denoting what can be ordered ('orderable') and the other what cannot ('not_orderable'), to meet the preferences of a group of individuals. Each person must find the selection satisfactory based on their likes and dislikes. The satisfaction criteria are: 1. A person is satisfied if at least one liked item is in 'orderable' list or one disliked item is in 'not_orderable' list. 2. No item can appear on both lists. 3. All participants must be satisfied by the combination of the two lists. 4. If no such combination exists that satisfies all, output empty lists for both. You always think step-by-step and show all your work in the explanation. Output your final solution as a comma-separated list of strings in Python code $\langle orderable = [...], not\_orderable = [...]\rangle$.

`# Input for a new problem`
**Preferences**: Jay: Likes nachos, ratatouille. Dislikes pie. Ada: Likes pie. Dislikes burger, ravioli. Zoe: Likes ravioli. Dislikes pie, burger. Arun: Likes ratatouille. Dislikes pie, nachos. Ula: Likes ratatouille. Dislikes ravioli, nachos. Ying: Likes nachos, ratatouille. Dislikes burger.

---

To assess the reasoning capabilities of LLMs, we analyze their performance on two variants of the 3-SAT problem. The first variant is the 3-SAT Decision problem, where the LLM acts as a solver and must determine whether a given 3-SAT problem is satisfiable. If the problem has a satisfiable assignment, the LLM should respond with "yes" and with "no" if it is unsatisfiable. The second variant is the 3-SAT Search problem, where the LLM's task extends beyond providing a simple "yes" or "no" response. If the formula is satisfiable, the LLM should also return an assignment to the variables that satisfies the formula. However, if the formula is found to be unsatisfiable, the LLM should once again respond with "no".

Due to the widespread adoption and best-in-class performance[5] we use GPT-4 Turbo (specifically, GPT-4 1106-preview model) as the main LLM reference to perform our experimental evaluation. However, we also report extended comparisons of GPT-4 to additional state-of-the-art LLMs, including both open-source (Llama 2 70B chat-hf (Touvron et al., 2023), Mixtral $8 \times 7$B (Jiang et al., 2024)) as well as proprietary models (GPT-3.5 Turbo (OpenAI, 2022), Gemini 1.0 Pro (Google, 2023), PaLM 2 text-bison (Anil et al., 2023)). The open-source models were run on 4 NVIDIA-DGX A-100 GPUs using a distributed (model parallel) framework. The generation configurations are listed in Appendix Table 2.

---

[5]https://huggingface.co/spaces/lmsys/chatbot-arena-leaderboard

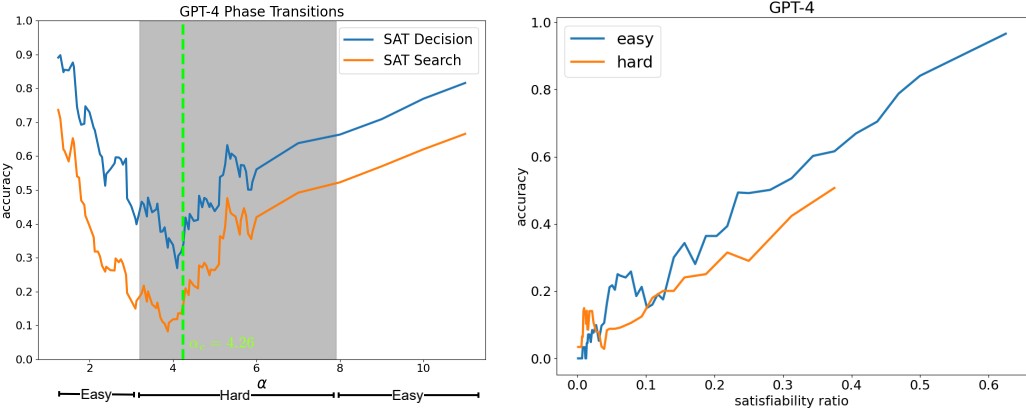

Figure 3: **Accuracy of GPT-4 against $\alpha$ and the satisfiability ratio.** [Left] Accuracy of GPT-4 against $\alpha$ for both SAT Decision and SAT Search. Notably, GPT-4's accuracy for both variants exhibits a significant decline around the critical point ($\alpha_c \approx 4.267$) aligning with the hard region. The dip in the performance mimics the increased solving time of the MiniSAT solver in the hard region. The setting is SAT-CNF. [Right] Accuracy of GPT-4 against the satisfiability ratio. We only include satisfiable instances and analyse problems from the hard and easy regions separately. Note that in our dataset, the 3-SAT problems exhibited maximum satisfiability ratios of approximately $0.4$ for the hard region and $0.62$ for the easy regions. The plots were generated using a size 4 moving window on $\alpha$ values.

To verify the LLM-generated solutions for both SAT-Menu and SAT-CNF, we employed MiniSAT v2.2 (Eén & Sörensson, 2003) solver.

## 4.2 Dataset Generation

To generate 3-SAT data, we varied $\alpha = m/n$ as a parameter to guide the generation process. For each $n \in [1, 10]$, we selected $\alpha \in [1, 11]$ based on feasible values of $m$. The full range of values is provided in Table 1 in the Appendix. Each instance is labeled either as satisfiable or unsatisfiable using MiniSAT v2.2 (Eén & Sörensson, 2003). Additionally, each instance is annotated with model count (i.e. number of feasible solutions for a formula) using the D4 model counter (Lagniez & Marquis, 2017).

For SAT-Menu setup, we map each instance (i.e. CNF formula) to a menu selection puzzle. The goal is to select a combination of orderable and non-orderable food items in a way that satisfies everyone's preferences. To this end, a food item is sampled without replacement corresponding to the list of variables in the formula. Then, every clause in the formula is treated as the preferences for an individual, leading to the creation of two distinct lists for each person: "Likes," for food items linked to positive literals, and "Dislikes," for those associated with negated literals. This approach is exemplified in Box 2.

Our dataset consists of 60,000 formulas, among which 39,909 are satisfiable (SAT) and the remaining 20,091 are unsatisfiable (unSAT) instances. For our experiments, we randomly selected approximately 6,000 samples from this pool. Detailed statistics of our data distribution are provided in Appendix A.

## 5 Results

### 5.1 Can LLMs solve 3-SAT problems?

We evaluate GPT-4's performance by measuring its accuracy (of solving for SAT Search and prediction for SAT Decision) across formulas with varying $\alpha$. We present these results in Figure 3 [Left]. GPT-4 demonstrates an apparent reasoning competence in the easy regions, while its accuracy sig-

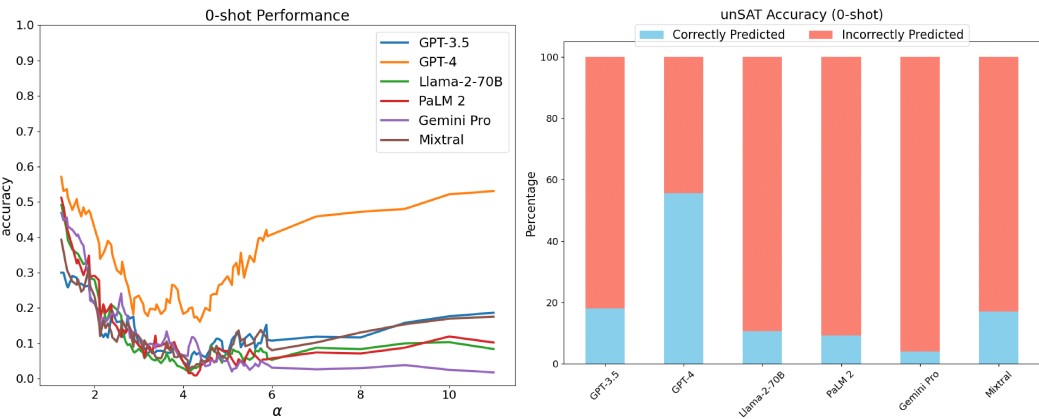

Figure 4: **Comparing GPT-4 with SOTA LLMs.** [Left] Phase transition characteristics for all LLMs. [Right] Comparison of unSAT accuracy (i.e. accuracy of correctly predicting unsatisfiable problems) on 3-SAT decision problem. GPT-4 outperforms all other models in detecting unsatisfiable problems. Both plots are on the search version of SAT-Menu setup with zero-shot prompting. The left plot was generated using a size 4 moving window on $\alpha$ values.

nificantly drops to $\approx 10\%$ in the hard region for SAT Search. We also observe that SAT Search poses a slightly greater challenge for GPT-4 than SAT Decision.

In Figure 3 [Right] we also plot GPT-4's performance against the *satisfiability ratio*, defined as $\frac{\text{model count}}{2^n}$, where model count is the number of satisfying assignments and $n$ is the number of variables. This ratio denotes the probability that a randomly selected variable assignment satisfies the given 3-SAT theory[6]. We can observe a clear dependence between the accuracy and the satisfiability ratio: formulas with more satisfying assignments tend to be easier for GPT-4 to solve. This holds across both easy and hard regions. Similar results were observed for other LLMs, as shown in Figure 7 in the Appendix.

In Figure 4 [Left], we can see that GPT-4 is the only model that exhibits solver-like phase transitions in the SAT Search problem. In contrast, other LLMs show a clear decline in performance in the high $\alpha$ region, displaying an Easy-Hard-Hard pattern. This is further supported in Figure 9 in the Appendix where we compare the phase transition characteristics across LLMs on both decision and search variants. While all LLMs perform significantly better on the decision variant, only GPT-4 consistently demonstrates an Easy-Hard-Easy phase transition in both. Moreover, GPT-4 is more accurate in detecting unsatisfiable instances, as shown in Figure 4 [Right] (Figure 11 in Appendix).

Finally, we explored in-context learning using three chain-of-thought input/output (I/O) demonstrations (Brown et al., 2020). These were randomly selected from a set of correctly solved problem instances (by the LLM) and manually checked for consistency between solutions and their explanations. Each I/O example included the input and the output solution, along with its chain-of-thought explanation. From Figure 10 in Appendix, we observe performance gains in the initial Easy-Hard phase but note a decrease in the subsequent Easy phase.

Note, that in all the above experiments, the performance of LLMs remains low in the Hard region.

As an ablation study, we attempted to empirically explore the claims that $T$ chain-of-thought steps can enhance the sequential reasoning abilities of a fixed-size transformer (Li et al., 2024) beyond $TC^0$. Based on this theory, one would expect the number of generated tokens (including that of CoT) to increase in the Hard region. As shown in Figure 12 in Appendix, GPT-4 appears to follow this expected pattern by generating more tokens in the Hard region, despite the performance drop. Consistent with our findings, other LLMs do not.

---

[6]Note that this is different from the probability that at least one satisfying assignment exists – two formulas can both be satisfiable but have different model counts and therefore different satisfiability ratios.

## 5.2 CAN AN LLM + SOLVER BOOST PERFORMANCE?

To aid LLMs in reasoning tasks, recent studies have explored pipelined approaches using LLMs to parse inputs into solver-compliant outputs, leveraging off-the-shelf solvers to derive the final answer (Ye et al., 2024; Liu et al., 2023). This approach is aligned with neurosymbolic techniques (De Raedt et al., 2020), which combine the universal function approximation capabilities of neural networks with the precision of symbolic systems.

To explore a similar setting in the context of 3-SAT, we ask whether we can augment LLMs with an external solver wherein the LLM translates (pseudo)-natural language into a format that a symbolic SAT solver, such as MiniSAT, can process. To this end, we prompt the LLM to translate 3-SAT formulas, which we provide in the SAT-Menu input format, into solver-compliant 3-SAT formulas. We then use a 3-SAT solver to solve the translated instance (see Box 4 in Appendix). We dub this approach **SAT-Translate** and plot its performance in Figure 5.

We observe that when LLMs have access to an external solver, there is a significant increase in their accuracy, reaching at best, in GPT-4's case, $\approx 100\%$ across the entire range of $\alpha$. We attribute this to the relatively lower computational complexity of translating 3-SAT formulas compared to solving them (i.e. finding satisfying assignments). Interestingly, we find that varying the input format between SAT-CNF and SAT-Menu does not significantly enhance LLMs inherent reasoning capabilities. The marked improvement in performance is primarily observed when they are equipped with an external solver. While ours is a straightforward approach, one could also explore tighter integrations as proposed in LLM-Modulo frameworks (Kambhampati et al., 2024a) which augments LLMs with critics and verifiers (Lightman et al., 2024; Hazra et al., 2024b), recognizing the ability of LLMs as *approximate* idea-generators for problems as against directly solving them..

## 6 DISCUSSION

At first sight, our experimental evaluation can be interpreted as indicating that LLMs, specifically GPT-4, exhibit a certain degree of reasoning capabilities: in the easy regions, GPT-4 solves some 3-SAT problems. One might then argue that GPT-4 does not reach perfect accuracy in these regions because of the scale of the model, and that simply increasing this will resolve the issue. For the hard region, however, it is unlikely that scaling up the model will result in radical improvements.

To explain this, reconsider Figures 2 and the time vs. $\alpha$ plot specifically. The reason why MiniSAT is capable of solving problems in the easy regions faster than problems around $\alpha_c$ is due to the heuristics built into the solver that guide the search for satisfying solutions (e.g. unit propagation and clause learning (Marques-Silva & Sakallah, 1999)). That is, heuristics work well when they can exploit statistical features in the problem instance to be solved. To date, there are no known heuristics that work well around $\alpha_c$ (and they are unlikely to exist due to the NP-hardness of 3-SAT). Solvers therefore have to resort to brute force search around $\alpha_c$.

In this light, we can reinterpret the experimental performance of GPT-4 on 3-SAT problems: GPT-4's apparent reasoning capabilities (in the easy regions) is due to the presence of statistical features that it can leach onto. For instance, GPT-4 may oversimplify and deem a problem unsatisfiable due to the high number of input tokens, which often works for overconstrained formulas (see Appendix B). Conversely, in the hard region, the drop in performance can be attributed to GPT-4's – and by extension current transformer-based LLMs' – inability to reason according to Bottou's definition. A similar observation has been made for a computationally tractable class of problems (not NP-complete) using BERT-style language models (Zhang et al., 2023).

Even though the basic random 3-SAT phase transition provides a rigorous framework for studying the reasoning capabilities of LLMs, we need to point out that there exists many more results about 3-SAT that might be considered as well. For instance, it has been found that certain aspects of the hardness of 3-SAT instances cannot be explained using the easy-hard-easy regimes induced by the phase transition (Mu & Hoos, 2015). Nevertheless, the phase transition seems to be adequate in our settings as demonstrated by the drop in performance of LLMs in the vicinity of $\alpha_c$. It should, however, be noted that 3-SAT instances encountered in practice often fall into the easy regions. Therefore, decision problems of interest do not usually exhibit worst-case complexity.

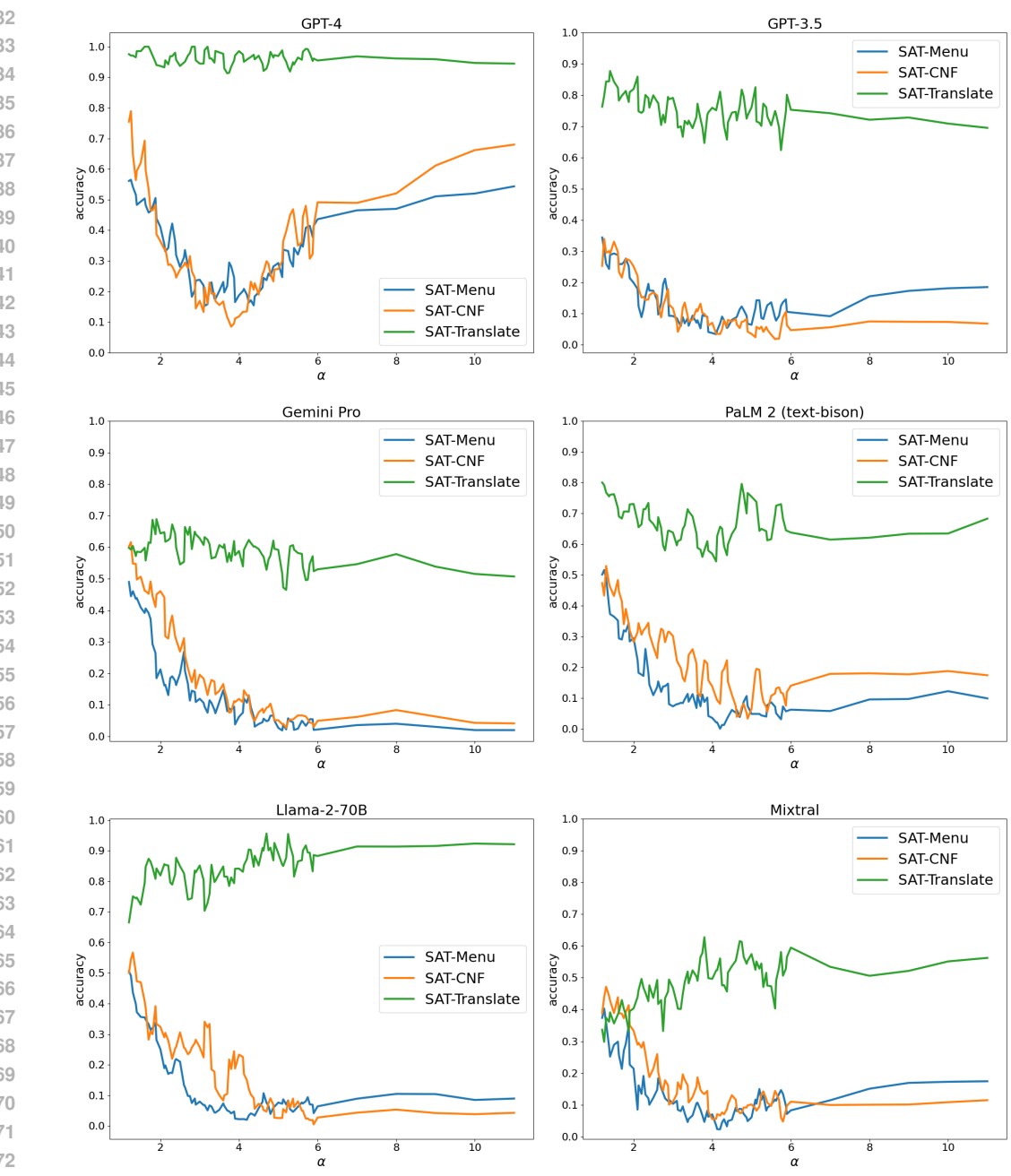

Figure 5: **LLM+Solver Framework comparison.** We compare the SAT Search performance of LLM+Solver frameworks (SAT-Translate) with standalone LLM on SAT-Menu and SAT-CNF variants. Remarkably, the SAT-Translate approach (plotted in green) outperforms the rest, showing the significance of augmenting LLMs with symbolic solvers. The plot was generated using a size 4 moving window on $\alpha$ values.

One could also argue that phase transitions are present in 2-SAT and several subclasses of Horn-SAT[7] motivating an analogous empirical evaluation. Accordingly, we examined GPT-4's performance on 2-SAT, 1-2-HornSAT, and 1-3-HornSAT problems. However, random 2-SAT, which can

---

[7]In particular, Demopoulos & Vardi (2006) explored $j - k$ Horn formulas, which consist of clauses that are either of length $j$ or $k$, and $j < k$. Here, the length of the formulas are $O(n^k)$.

be solved in polytime, only exhibits a barely detectable phase transition (Chvatal & Reed, 1992; Goerdt, 1996). We observed the same with GPT-4 as shown in Figure 13 where it performed reasonably well for both the Search and Decision variants.

In contrast, for Horn-SAT, we observed a notable performance drop for GPT-4 around the satisfiability threshold – the point where the probability of satisfiability transitions from $> 0.5$ to $< 0.5$ – highlighted in Figure 14. This suggests that while GPT-4 performs robustly on NL-complete problems like 2-SAT, its effectiveness diminishes for higher complexity classes such as P-complete (Horn-SAT) and NP-complete (3-SAT). These observations align with the findings of Peng et al. (2024); Li et al. (2024) which suggest that multi-layer transformers cannot solve problems such as Derivability, 2-SAT, Horn SAT, and Circuit Evaluation unless L=NL. However, with $T$-CoT steps (where $T$ scales polynomially with sequence length), can compute any function solvable by a polynomial-sized circuit. That said, each of these SAT fragments presents interesting structural and algorithmic properties. Therefore, they represent distinct avenues of exploration, each requiring comprehensive and focused work due to their unique structural and algorithmic properties. Undertaking such an expansive analysis exceeds the scope of this study.

## 7 CONCLUSION

A superficial analysis of the reasoning capabilities of LLMs suggests they possess strong and complex reasoning capabilities – a common fallacy (Bubeck et al., 2023). However, our detailed experimental analysis indicates that what appears as reasoning capabilities could be a mirage, with LLMs predominantly exploiting statistical features present in the data and would explain why LLMs have been termed "statistical parrots" (Bender et al., 2021). These findings are supported by our experiments on lower complexity classes (2-SAT and fragments of Horn-SAT), where GPT-4 performed comparatively well on NL-complete problems but rather poorly on P-hard problem. We believe that the comprehensive experimental protocol we established for 3-SAT can be deployed in future studies to further investigate the reasoning capabilities of LLMs – at a more fine-grained level than worst-case theoretical studies.

This is not to say that LLMs lack value – quite the opposite. LLMs are highly effective at translating problems into a formal language and then passing these problems on to solvers. This utility is demonstrated by the relatively superior performance of a straightforward LLM+Solver framework. This suggests that effective reasoning with LLMs should involve decomposing tasks into easy LLM problems and hard symbolic problems (when possible), rather than solely relying on scaling models with more training data and compute for natural language reasoning. In a more general context, it would require that the class of the problem is recognized correctly – here 3-SAT – and that the right solvers are available.

Furthermore, LLMs can serve as valuable knowledge bases, leveraging their extensive common sense and world knowledge through natural language queries to guide search processes. This can be utilized by either solvers guiding LLMs (Hazra et al., 2024b) or vice versa, where LLMs assist solvers (Wong et al., 2023).

## REPRODUCIBILITY

We provide comprehensive instructions to ensure the reproducibility of the results. The dataset generation process and statistics are outlined in § 4.2 and Appendix A, the LLM configurations are listed in Appendix Table 2, and detailed prompts shown in Appendix C. Moreover, we also discuss limiting/failure cases in Appendix B and how they can be mitigated. We intend to open-source our code and data following publication. The code base will be made available to the reviewers through an anonymous link.

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

# APPENDIX

The appendix is organized as follows. Dataset Statistics § A, Output analysis of LLMs § B, and Full Prompts § C.

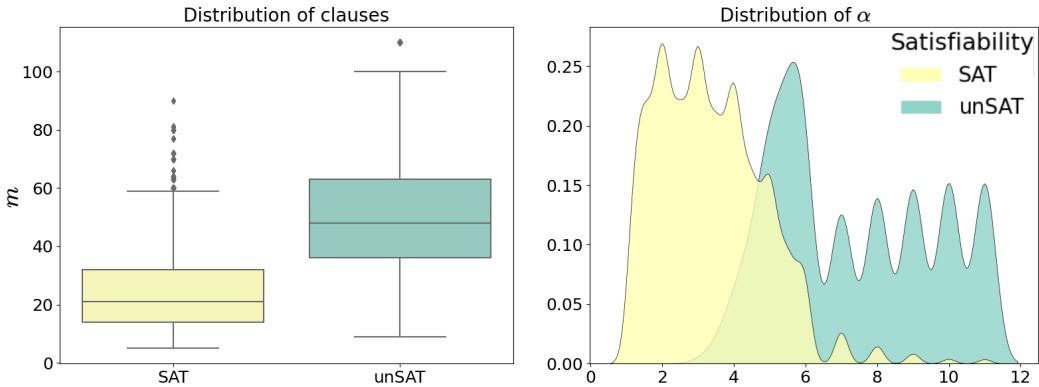

Figure 6: **3-SAT Dataset Statistics**. Figures depict clauses $m$ (left) and $\alpha$ (right) distribution across SAT and unSAT instances, highlighting that unSAT problems typically feature more clauses and higher $\alpha$ values.

## A  DATASET STATISTICS

### A.1  3-SAT

Table 1 lists all values of the range of $\alpha$ corresponding to each $n$. For $\alpha$ values within the $(6, 11]$ interval, we incremented $\alpha$ by 1. For the $[1, 6]$ interval, which contains the most "interesting problems," we aimed for finer granularity by choosing the smallest possible $\alpha$ increment. This increment ensures that, given the number of variables $n$, we obtain an integer number of clauses $m$. For example, with $n = 3$, the minimum increment is $\alpha_{inc} = 1$, and for $n = 4$, it is $\alpha_{inc} = 0.25$. For each $\alpha_{inc}$ we generated 300 formulas.

The distribution of formulas according to the number of variables is detailed as follows: 3,000 formulas with 3 variables, 7,500 with 4 variables, 9,000 with 5 variables, 4,500 with 6 variables, 3,000 with 7 variables, 13,500 with 8 variables, 3,000 with 9 variables, and 16,500 with 10 variables.

As shown in Figure 6, the range of clauses ($m$) varies in [5, 90], and the alpha ($\alpha = m/n$) ranges in [1.1, 11.0]. For unSAT instances, the clauses range from 9 to 110, with the alpha varying in [2.60, 11.0]. Across the entire dataset, the average number of variables ($n$) is 7.2, the average number of clauses ($m$) is 33, and the mean $\alpha$ is 4.7. This diverse dataset provides a broad spectrum for analyzing the impact of variable and clause distribution on formula satisfiability.

### A.2  2-SAT

For the experiments on 2-SAT, we followed the same formula generation procedure used for 3-SAT. However, the $\alpha$ range was reduced to $[1, 10]$, as the unsatisfiability transition occurs earlier in 2-SAT problems. This adjustment allowed us to reduce the dataset size while preserving relevant data for analysis. For each $\alpha$ value, detailed in Table 1, we generated 100 formulas. The resulting dataset comprises a total of 29,600 formulas, of which 4,860 are satisfiable.

### A.3  HORN-SAT

We generated two classes of random Horn SAT formulas, namely 1-2-HornSAT and 1-3-HornSAT, following the methodology outlined by Moore et al. (2007). Intuitively, 1-2-HornSAT formulas

Table 1: Table shows the range of alpha value for each $n$ (i.e. number of variables) in the generated dataset. We generate $300$ formulas per $\alpha$ value.

| $n$ | Range of $\alpha$ |
|---|---|
| 3 | 1.0, 2.0, 3.0, 4.0, 5.0, 6.0, 7.0, 8.0, 9.0, 10.0, 11.0 |
| 4 | 1.0, 1.25, 1.5, 1.75, 2.0, 2.25, 2.5, 2.75, 3.0, 3.25, 3.5, 3.75, 4.0, 4.25, 4.5, 4.75, 5.0, 5.25, 5.5, 5.75, 6.0, 7.0, 8.0, 9.0, 10.0, 11.0 |
| 5 | 1.0, 1.2, 1.4, 1.6, 1.8, 2.0, 2.2, 2.4, 2.6, 2.8, 3.0, 3.2, 3.4, 3.6, 3.8, 4.0, 4.2, 4.4, 4.6, 4.8, 5.0, 5.2, 5.4, 5.6, 5.8, 6.0, 7.0, 8.0, 9.0, 10.0, 11.0 |
| 6 | 1.0, 1.5, 2.0, 2.5, 3.0, 3.5, 4.0, 4.5, 5.0, 5.5, 6.0, 7.0, 8.0, 9.0, 10.0, 11.0 |
| 7 | 1.0, 2.0, 3.0, 4.0, 5.0, 6.0, 7.0, 8.0, 9.0, 10.0, 11.0 |
| 8 | 1.0, 1.125, 1.25, 1.375, 1.5, 1.625, 1.75, 1.875, 2.0, 2.125, 2.25, 2.375, 2.5, 2.625, 2.75, 2.875, 3.0, 3.125, 3.25, 3.375, 3.5, 3.625, 3.75, 3.875, 4.0, 4.125, 4.25, 4.375, 4.5, 4.625, 4.75, 4.875, 5.0, 5.125, 5.25, 5.375, 5.5, 5.625, 5.75, 5.875, 6.0, 7.0, 8.0, 9.0, 10.0, 11.0 |
| 9 | 1.0, 2.0, 3.0, 4.0, 5.0, 6.0, 7.0, 8.0, 9.0, 10.0, 11.0 |
| 10 | 1.0, 1.1, 1.2, 1.3, 1.4, 1.5, 1.6, 1.7, 1.8, 1.9, 2.0, 2.1, 2.2, 2.3, 2.4, 2.5, 2.6, 2.7, 2.8, 2.9, 3.0, 3.1, 3.2, 3.3, 3.4, 3.5, 3.6, 3.7, 3.8, 3.9, 4.0, 4.1, 4.2, 4.3, 4.4, 4.5, 4.6, 4.7, 4.8, 4.9, 5.0, 5.1, 5.2, 5.3, 5.4, 5.5, 5.6, 5.7, 5.8, 5.9, 6.0, 7.0, 8.0, 9.0, 10.0, 11.0 |

contain clauses with only 1 and 2 literals, while 1-3-HornSAT formulas contain clauses with only 1 and 3 literals. These classes correspond to 2-SAT and 3-SAT in the Horn category, which are P-complete rather than NP-complete.

Specifically, we sampled formulas from $H^2_{n,d_1,d_2}$ and $H^3_{n,d_1,0,d_3}$ distributions, where $d_1$ was fixed to 0.5 to simplify the sampling space, and $d_2$ and $d_3$ (referred to as $\alpha$) were varied to analyze formula behavior. For consistency with 2-SAT and 3-SAT analyses, $\alpha$ was incremented from 0 to 12 in steps of 1, resulting $m = \alpha n + 0.5n + 1$ clauses for these Horn formulas. Notice that for $\alpha = 0$, the formulas still contain $0.5n + 1$ clauses due to the fixed $d_1$ parameter. The choice of $d_1 = 0.5$ ensures we can observe a satisfiability threshold across different $\alpha$ values and small $n$. Lower values produced trivially satisfiable formulas at small $n$, for all $\alpha$ values, and were therefore avoided. We empirically explored $d_1$ values starting at 0.2 in increments of 0.1 before selecting 0.5.

We generated 300 formulas for each $\alpha$ value and formula type, with $n$ ranging from 3 to 10. This range aligns with our other experimental settings, as opposed to the significantly larger $n = 20,000$ used by Moore et al. (2007). Each dataset contains 31,200 formulas. Among these, 5,036 formulas are satisfiable in the 1-3-HornSAT class, and 3,898 are satisfiable in the 1-2-HornSAT class.

## B  LLMs as Solvers: Output Analysis

We observed the following behaviors in the generated outputs, including chain-of-thought (CoT) reasoning:

**Diverse Reasoning Techniques**: GPT-4 employs varying reasoning techniques depending on the prompt type (SAT-CNF vs. SAT-Menu) and even adapts its approach across individual problems within the same prompt type.

**SAT-CNF Reasoning**: The dominant strategy involves backtracking, as illustrated in Box 5. Occasionally, GPT-4 employs local search, where it assigns items to "orderable" and "not-orderable" lists and iteratively modifies these based on detected conflicts (e.g., *We can create two sets for liked and disliked items and then compare them to find any conflicts. Let's begin by creating a list of all the likes and dislikes to identify conflicts.*).

**SAT-Menu Reasoning**: The primary strategy here is trial-and-error. Occasionally, GPT-4 applies heuristics such as the Maximum Occurrence in Minimum-sized clauses (MOM) heuristic to prioritize variables appearing most frequently in the smallest clauses (e.g., *We start by making a tally of how many people like or dislike each food item... If we put 'macaron' on the 'orderable' list, we will satisfy many people who like it.*).

Table 2: Configuration Parameters

| Parameter | Value |
|---|---|
| GPT-* | |
| temperature | 1 |
| max_tokens | 4096 |
| top_p | 1 |
| frequency_penalty | 0 |
| presence_penalty | 0 |
| Llama 2 70B | |
| 4-bit quantization | True |
| tokenizer type | Fast |
| max_new_tokens | 2048 |
| batch_size | 20 |
| Mixtral $8 \times 7$B | |
| 4-bit quantization | True |
| attention | Flash |
| max_new_tokens | 4096 |
| batch_size | 30 |

**"Lazy" Solutions**: As noted in subsequent text, GPT-4 often produces "lazy" solutions in many cases, either providing an outline of how to solve the problem or asking to be delegated to a solver.

We also discuss some interesting failure cases we observed during our experiments. In the SAT-CNF context, it was observed that LLMs, including GPT-4, often opt to pass the task to an external SAT solver rather than solving it themselves. Additionally, when attempting to find a solution, these models tend to provide only a conceptual outline of the solution instead of a concrete answer, a tendency that becomes more pronounced with larger formulas. When prompted explicitly for a solution, GPT-4 might simplistically conclude that the problem is unsatisfiable due to its complexity, as shown in Box 5. Although this reasoning is not entirely sound – as over-constrained formulas can still potentially be solvable – it appears that LLMs might be leveraging this as a statistical feature.

In the SAT-Translate approach, it was observed that LLMs (with the exception of GPT-4) frequently deviated from the specific requirements of the prompt, generating solutions that did not adhere to the syntax expected by the solver. These inconsistencies in translation led to suboptimal accuracy levels within the LLM+Solver frameworks.

It should be noted that the input to the LLM is based on fixed templates, the rules of which can be captured using regular grammar. Thus, one could write a simple parser to map the menu input to a CNF formula. However, mapping from ambiguous natural language to solver-compliant input may be non-trivial for the LLMs. Generally, outputs from LLMs often require additional post-processing to meet specific guidelines.

## C  FULL PROMPTS

Here, we provide full prompts used in our experiments for SAT-Menu (Box 2), SAT-CNF (Box 3), and SAT-Translate (Box 4).

---

**Box 2: SAT-Menu Prompt**

# System Message
Your task is to output two distinct lists of food items, one denoting what can be ordered ('orderable') and the other what cannot ('not_orderable'), to meet the preferences of a group of individuals. Each person must find the selection satisfactory based on their likes and dislikes. The satisfaction criteria are: 1. A person is satisfied if at least one liked item is in 'orderable' list or one disliked item is in 'not_orderable' list. 2. No item can appear on both lists. 3. All participants must be satisfied by the combination of the two lists. 4. If no such combination exists that satisfies all, output empty lists for both. You always think step-by-step and show all your work in the explanation. Output your final solution as a comma-separated list of strings in Python code $\langle orderable = [...], not\_orderable = [...]\rangle$.

# Pairs of Preferences and Solutions for in-context learning
**Few-shot Examples** . . .

# Input for a new problem
**Preferences**: Jay: Likes nachos, ratatouille. Dislikes pie. Ada: Likes pie. Dislikes burger, ravioli. Zoe: Likes ravioli. Dislikes pie, burger. Arun: Likes ratatouille. Dislikes pie, nachos. Ula: Likes ratatouille. Dislikes ravioli, nachos. Ying: Likes nachos, ratatouille. Dislikes burger.

# Generated Solution for the given problem
**Solution**: First, let's list out the likes and dislikes of each person and then categorize the food items based on their preferences.

Jay: Likes nachos, ratatouille. Dislikes pie.
Ada: Likes pie. Dislikes burger, ravioli.
Zoe: Likes ravioli. Dislikes pie, burger.
Arun: Likes ratatouille. Dislikes pie, nachos.
Ula: Likes ratatouille. Dislikes ravioli, nachos.
Ying: Likes nachos, ratatouille. Dislikes burger.

Now, let's categorize the items as 'orderable' and 'not_orderable' based on the given preferences.

From the above preferences, we can create the following lists:

Orderable: pie, ratatouille, nachos
Not orderable: burger, ravioli

Let's examine if this combination satisfies all the participants' preferences.

1. Jay: Satisfied by ratatouille in orderable list.
2. Ada: Satisfied by pie in orderable list.
3. Zoe: Satisfied by burger in not_orderable list.
4. Arun: Satisfied by ratatouille in orderable list.
5. Ula: Satisfied by ratatouille in orderable list.
6. Ying: Satisfied by nachos in orderable list.

All participants are satisfied with this combination, and no item appears in both lists.
So the final lists are:

```python
orderable=[pie, ratatouille, nachos]
not_orderable=[burger, ravioli]
```

---

Box 3: SAT-CNF Prompt

# System Message
Let's play the SAT (satisfiability) game. The input is a list of clauses, where each clause is represented as a disjunction of literals (variables or their negation connected by logical OR). Your task is to find valuation of Boolean variables such that a Boolean CNF formula evaluates to True. The solution should be in form of a dictionary where keys are variables and values are Boolean (True or False). The satisfaction criteria are: 1. At least one literal in each clause should be True. 2. A variable can't be both True and False in the dictionary. 3. If no satisfying assignment exists, you should output an empty dictionary. You always think step-by-step and show all your work in the explanation. Output the solution in Python code dictionary, enclosed within $\langle output : ... \rangle$.

# Input for a new problem
**Formula**:

$$[[-3, 1, -4], [-4, -2, 1], [-1, -4, 5], [5, 1, 2], [-5, 4, 2], [-4, 3, 1],$$
$$[1, 5, -3], [-2, 1, 3], [1, -5, -4], [4, -3, -1], [-2, 5, -3]]$$

# Generated Solution for the given problem
**Solution**: To find a satisfying assignment for this CNF (Conjunctive Normal Form) formula, we will use a simple approach and try to construct a solution iteratively while checking at each step if the clauses are satisfied or not. We will go through the clauses one by one and make arbitrary decisions on the truth values of the literals. We have to ensure that no variable is assigned both True and False, and each clause must have at least one True literal.

We start with the first clause [-3, 1, -4]. Let's arbitrarily decide:
3 is False (so -3 is True),
1 is True (so we satisfy the clause [-3, 1, -4] because 1 is True or -3 is True),
4 is True (since we haven't made any decision on 4, we do not yet satisfy the clause with -4, but that won't be an immediate issue).

Our current assignments are now: 1: True, 3: False, 4: True.

Next, we consider the second clause [-4, -2, 1]. Since 1 is True, this clause is already satisfied.

Next is [-1, -4, 5]. We need to satisfy this clause without changing the truth value of 1 or 4, so we choose to make 5 True.

Our assignments are now: 1: True, 3: False, 4: True, 5: True.

Next, we look at [5, 1, 2]. Since 5 is True, this clause is already satisfied.

Next is [-5, 4, 2]. We need to satisfy this without changing 5 or 4, so we set 2 to True.

Our current assignments are: 1: True, 2: True, 3: False, 4: True, 5: True.

Now the next clause is [-4, 3, 1]. Since 4 is True, -4 is False, but 1 is True, so the clause is satisfied.

For the clause [1, 5, -3]. This clause is already satisfied by 1 being True.

For [-2, 1, 3]. This clause is also satisfied because 1 is True.

For [1, -5, -4]. This is again satisfied because 1 is True.

Next clause is [4, -3, -1]. This clause is satisfied with 4 being True.

The last clause is [-2, 5, -3]. This is satisfied because 5 is True.

All clauses are satisfied with our current assignments, so we return:

```python
output:  {1:  True, 2:  True, 3:  False, 4:  True, 5:  True}
```

Box 4: SAT-Translate Prompt

# System Message
You are provided with a list of preferences from different individuals, each specifying items they like and dislike. Create a logical expression in Conjunctive Normal Form (CNF) that satisfies a set of individual preferences regarding likes and dislikes of certain items. The condition for an individual's satisfaction is that either at least one item they like is included, or at least one item they dislike is excluded in your selection. Format the final CNF expression in LaTeX. Ensure all item names are retained in the final output. Do not include any explanation.

# Input for a new problem
**Preferences**: Om: Likes naan, curry. Dislikes tandoori. Bao: Likes curry. Dislikes naan, tandoori. Nic: Likes naan. Dislikes curry, tandoori. Pat: Likes curry. Dislikes naan, tandoori. Du: Likes tandoori, naan, curry. Kim: Likes curry. Dislikes tandoori, naan.

# Generated Solution for the given problem
**Solution**:

$(\text{naan} \lor \text{curry} \lor \neg\text{tandoori}) \land (\text{curry} \lor \neg\text{naan} \lor \neg\text{tandoori}) \land (\text{naan} \lor \neg\text{curry} \lor \neg\text{tandoori}) \land$

$(\text{curry} \lor \neg\text{naan} \lor \neg\text{tandoori}) \land (\text{tandoori} \lor \text{naan} \lor \text{curry}) \land (\text{curry} \lor \neg\text{tandoori} \lor \neg\text{naan})$

**Box 5: SAT-CNF (Lazy Solution)**

# System Message
Let's play the SAT (satisfiability) game. The input is a list of clauses, where each clause is represented as a disjunction of literals (variables or their negation connected by logical OR). Your task is to find valuation of Boolean variables such that a Boolean CNF formula evaluates to True. The solution should be in form of a dictionary where keys are variables and values are Boolean (True or False). The satisfaction criteria are: 1. At least one literal in each clause should be True. 2. A variable can't be both True and False in the dictionary. 3. If no satisfying assignment exists, you should output an empty dictionary. You always think step-by-step and show all your work in the explanation. Output the solution in Python code dictionary, enclosed within $\langle output : ... \rangle$.

# Input for a new problem
**Formula**:

$[[7, 9, -4], [1, -8, -4], [-1, 3, -8], [10, 3, -8], [9, -1, -8], [-4, 10, 6], [-7, 1, -8], [-3, 5, -6],$
$[-5, 8, 6], [8, -9, 5], [8, -5, -9], [5, -8, -9], [-2, 4, 9], [8, -2, 6], [9, -5, 6], [-10, -4, -2], [-1, 2, -8]$
$[-3, -5, -4], [-8, -2, -3], [1, 3, 9], [8, -10, -1], [-8, -2, -10], [4, -10, 3], [-9, 4, -8], [7, 8, -10],$
$[-4, 5, -3], [7, 6, 2], [-5, -7, -4], [-4, 7, -9], [-10, 2, 5], [3, -1, 10], [-5, -4, -7], [-3, 8, 4],$
$[-10, -3, -4], [6, 4, -1], [8, 7, 9], [1, 3, 10], [10, 8, 9], [-3, 9, 4], [-5, 9, -1], [-4, 8, 9], [2, -4, -1],$
$[-4, 10, -7], [5, -7, -10], [2, 8, 7], [9, -2, 10], [-7, 2, -4], [5, -3, 4], [1, -5, 4]]$

# Generated Solution for the given problem
**Solution**: To solve this satisfiability problem, we need to determine an assignment of truth values to the variables such that all clauses are satisfied. With such a large and complex problem, an efficient systematic approach or use of an algorithm like the DPLL (Davis–Putnam–Logemann–Loveland) algorithm would be ideal. However, since we're expected to think step-by-step, let's attempt a simplified analysis.

We start by looking for unit clauses, which contain only one literal, or pure literals, which appear in one polarity only throughout the formula. Observing the clauses, we can't immediately spot such helpful shortcuts due to the complexity. Instead, we can iterate through the clauses, and try to assign truth values to the variables in a way that satisfies as many clauses as possible. We might need to backtrack if we reach a contradiction.

We must make some initial choices and see if those choices can lead to a consistent solution. Let's start with the first clause '[7, 9, -4]'. We can begin by setting 7 to True, 9 to True, or the negation of 4 (-4) to True. Let's decide to set 7 to True. This is an arbitrary choice at the beginning, and we may need to backtrack if this leads to a contradiction later.

Now, we need to check other clauses and see if this choice is consistent with them and try to fill in further values. We will need to iterate through each clause and see if it's possible to satisfy it given our previous choices.

Given the complexity of the problem and the large number of variables and clauses involved, solving it by hand or by a step-by-step explanation is impractical. In practice, we would use a SAT solver software to find the satisfying assignment, if one exists. A SAT solver automatically handles the process of proposing variable assignments, backtracking, and deducing necessary assignments based on the structure of the formula.

For the purposes of this answer, since I cannot run a SAT solver in real-time, I will have to state that solving this problem by hand is unfeasible within this format. The output, given the complexity, cannot be determined without the use of algorithmic assistance. If no satisfying assignment exists (meaning the formula is unsatisfiable), or we cannot solve it by a simple theoretical process within this framework, the output would be an empty dictionary.

```python
output:  {}
```

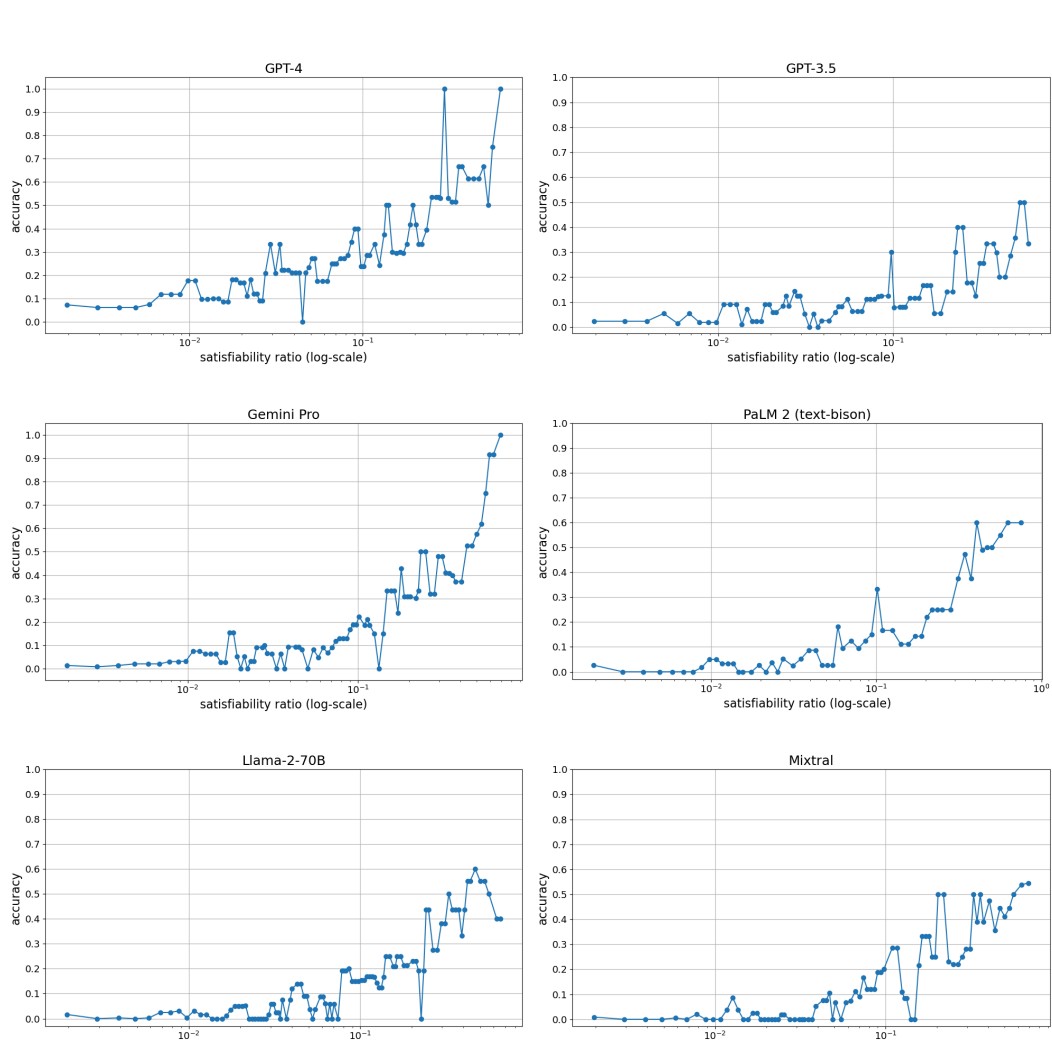

Figure 7: **LLM accuracy vs. satisfiability ratio.** The figure shows how accuracy improves with satisfiability ratio (in log-scale) across all LLMs. The setup used is the search version of SAT-Menu with 0-shot prompting.

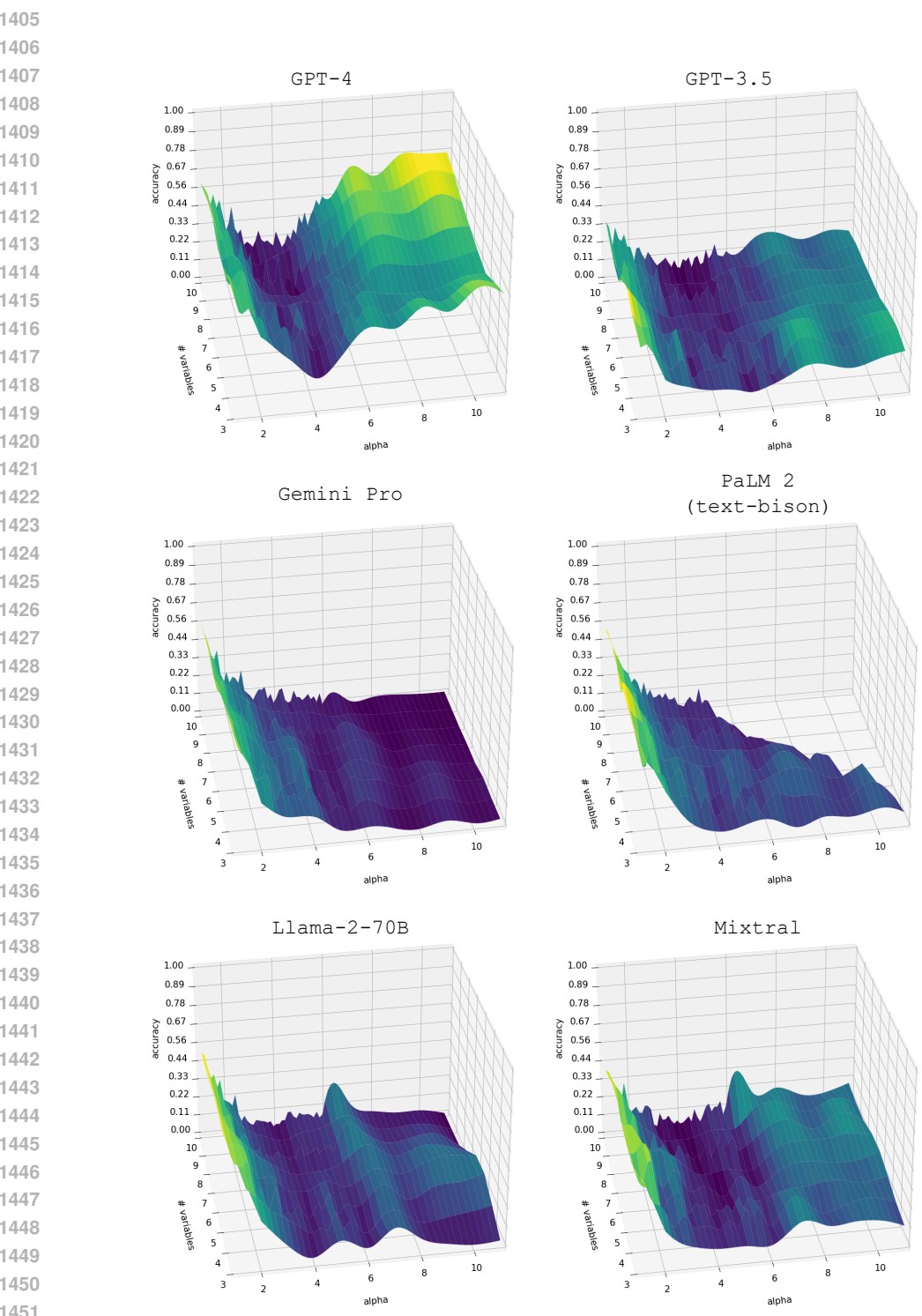

Figure 8: **LLM Phase Transitions.** Surface plots depict phase transition characteristics, comparing the accuracy in the range $[0, 1]$ against # variables and $\alpha$. The setup is the search version of SAT-Menu with zero-shot prompting. Unlike solvers, the surface plot does not show steepness with an increase in # variables.

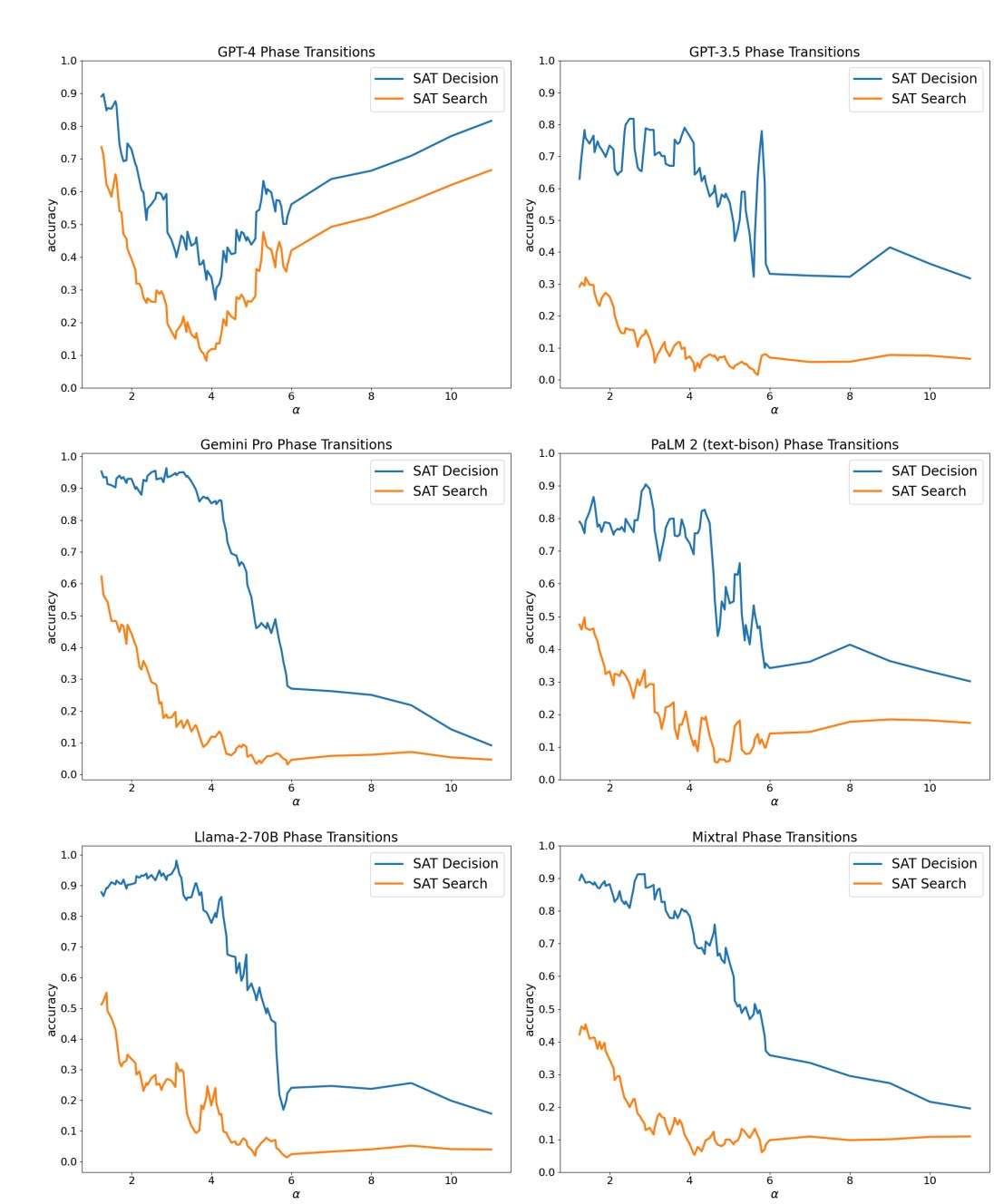

Figure 9: **SAT Decision vs. SAT Search comparison.** The figure illustrates the phase transitions in solving random 3-SAT problems, specifically comparing the decision and search variants. Although both variants fall under the same complexity class, empirical evidence shows that LLMs more readily solve the decision problem than the search problem. Among these models, only GPT-4 demonstrates the distinct Easy-Hard-Easy phase transition characteristic. The setup is SAT-CNF with 0-shot prompting. The plot was generated using a size 4 moving window on $\alpha$ values.

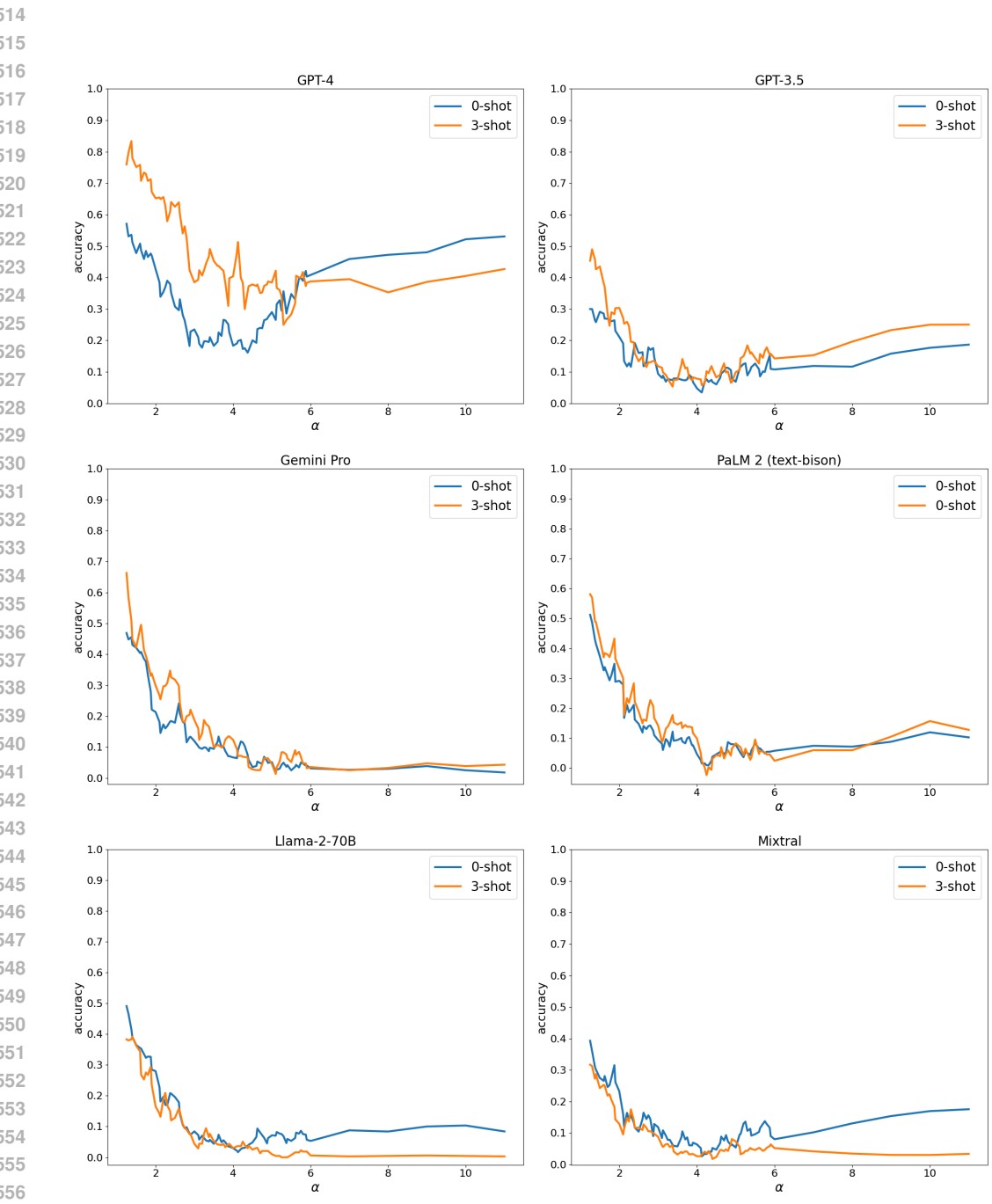

Figure 10: **0-shot vs. 3-shot accuracy.** The figure compares 0-shot and 3-shot performance comparing all LLMs. It can be observed that, except GPT-4, in-context learning does not enhance performance for other LLMs. The setup is the search version SAT-Menu. The plot was generated using a size 4 moving window on $\alpha$ values.

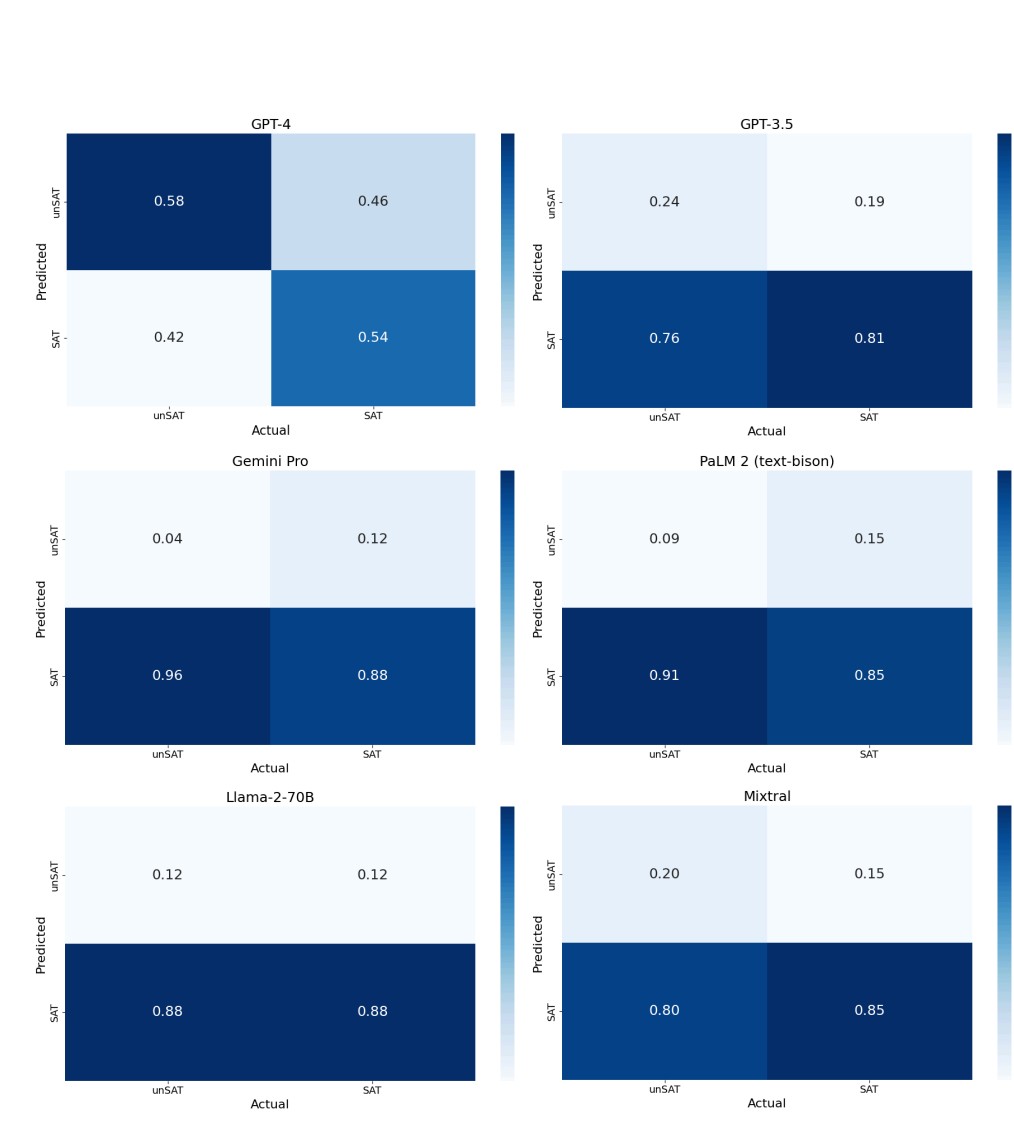

Figure 11: **Confusion matrices for 3-SAT Decision Problem.** The figure compares the accuracy of all LLMs on the 3-SAT decision problem. It can be observed that, except GPT-4, all other LLMs struggle to correctly classify unsatisfiable instances. The cell annotations reflect classification accuracy, normalized over the true counts (column) to account for the imbalance between SAT and unSAT instances. The setup is SAT-Menu with 0-shot prompting.

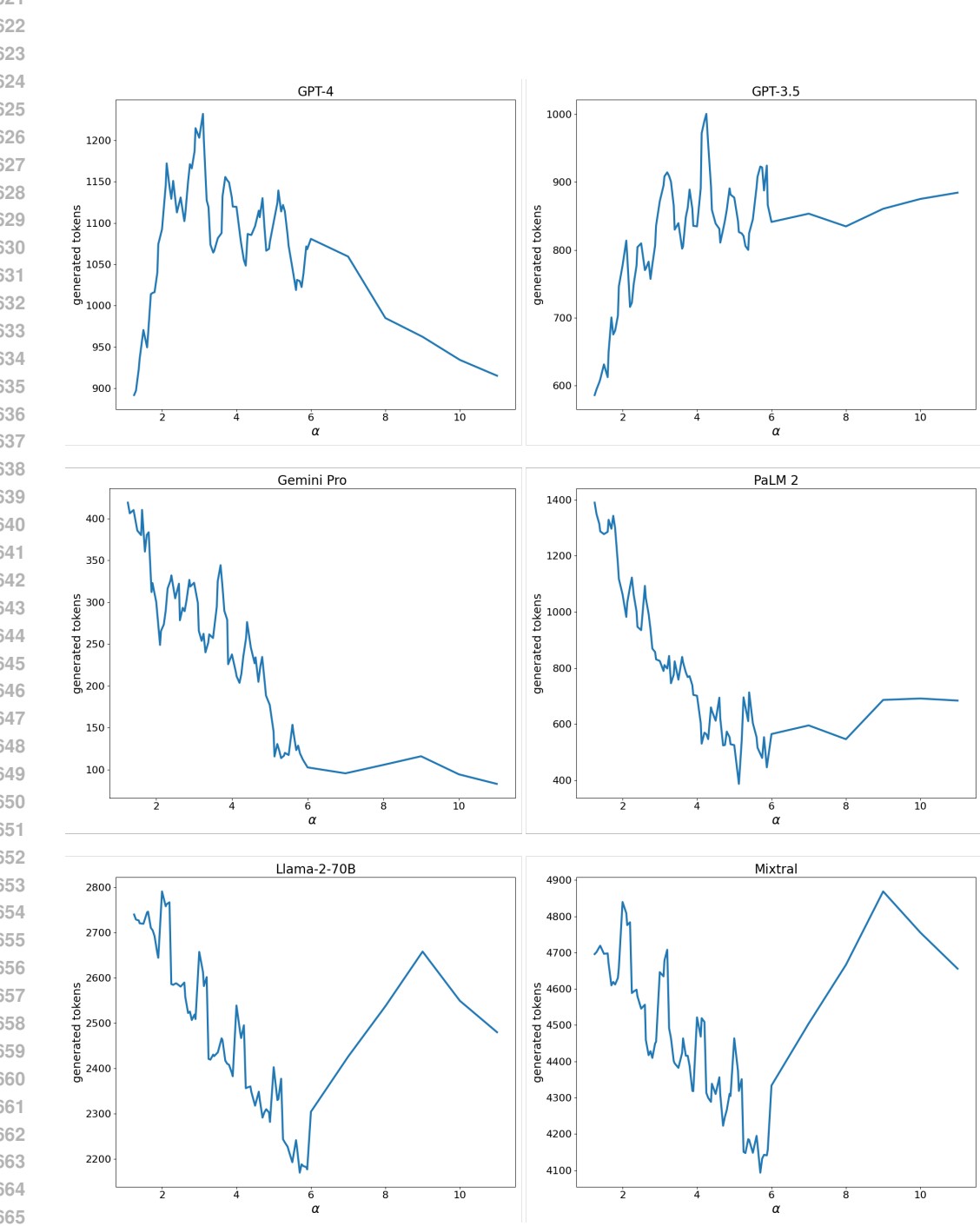

Figure 12: **# Generation tokens vs. $\alpha$.** The figure shows how the number of generated tokens varies with the hardness of the problem. The setup used is the search version of SAT-Menu with 0-shot prompting. The plot was generated using a size 4 moving window on $\alpha$ values.

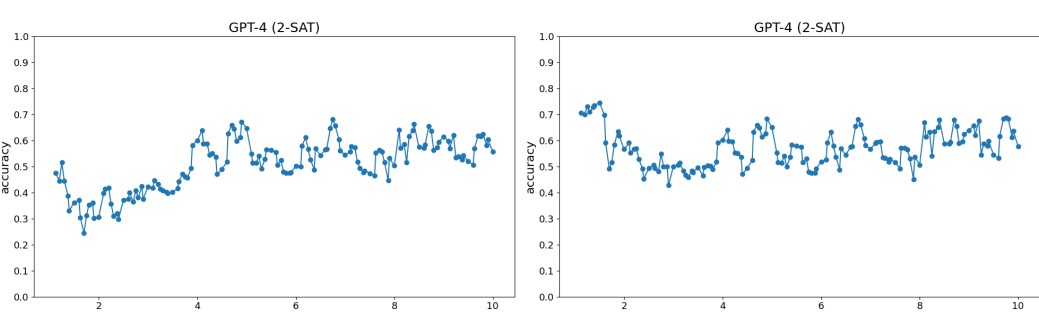

Figure 13: **SAT-Search [Left] and SAT-Decision [Right] performance on Random 2-SAT.** The figure illustrates GPT-4's performance on the search and decision variants of random 2-SAT. As expected, the phase transitions are less pronounced compared to 3-SAT, aligning with theoretical findings that phase transitions become detectable only for $k$-SAT for $k \geq 3$. The setting is SAT-Menu.

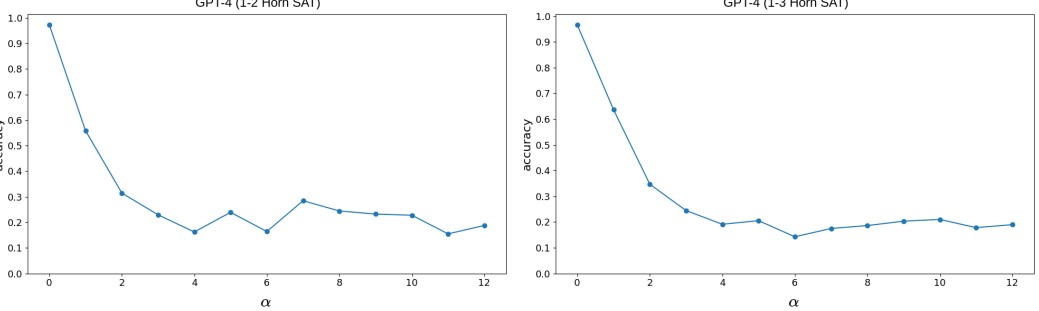

Figure 14: **SAT-Search performance on 1-2-HornSAT [Left] and 1-3-HornSAT [Right].** We observed a notable performance drop for GPT-4 around the satisfiability threshold – the point where the probability of satisfiability transitions from $> 0.5$ to $< 0.5$. The setting is SAT-CNF.

