# OpenReview forum: "Can Large Language Models Reason? A Characterization via 3-SAT"
_ICLR.cc/2025/Conference — Submitted to ICLR 2025_

### Official Review · Reviewer_pLV9 · 2024-10-29

**Soundness:** 2
**Presentation:** 3
**Contribution:** 2
**Rating:** 5
**Confidence:** 4

**Summary:**

This paper examines the reasoning capabilities of large language models (LLMs) through the perspective of statistical computational complexity. This framework analyzes computational complexity as a random variable influenced by specific order parameters of a given problem. The focus of this investigation is 3-SAT, a problem extensively studied in the AI literature for its phase transition phenomena. The authors demonstrate that state-of-the-art LLMs face significant challenges in solving random instances with up to 10 variables. Not only do LLMs struggle to resolve instances located in the phase-transition region, but they also often fail to address instances in the simpler regions that are either under-constrained or over-constrained. Additionally, the authors propose that an LLM-modulo SAT framework, which integrates an LLM architecture with a SAT solver, presents an alternative for tackling simple commonsense reasoning tasks that can be reframed as 3-SAT problems.

**Strengths:**

**Novelty:** While recent studies have examined the reasoning limitations of LLMs from a computational perspective, this paper provides new empirical findings through a statistical complexity analysis. One of the most noteworthy observations is the difficulty LLMs face in effectively addressing "simple" constraint satisfaction problems (CSPs). By "simple," I refer to CSP instances that involve no more than 10 variables, which are regarded as toy problems within the constraint programming community. For these instances, it appears that most LLMs do not significantly outperform a random baseline (Left part of Figure 4).

**Weaknesses:**

**Positioning:** Although I am convinced that the paper is conveying something new, the related work section could benefit from further elaboration, particularly in discussing the empirical results presented in this paper alongside recent studies that have theoretically and/or empirically explored the reasoning limitations of large language models (LLMs). Notably, Dziri et al. (2023) have experimentally demonstrated that the performance of Transformers on constraint reasoning tasks declines rapidly as the depth of the task increases. Furthermore, Peng et al. (2024) have shown that the satisfiability of Krom, Horn, or affine formulas cannot be determined by multi-layer Transformers unless L = NL. Hence, the new results concerning random 3-SAT instances are, at first glance, not particularly surprising.

**Clarity:** The first four sections are relatively clear, but I found the results in Section 5.1 and the significance of Section 5.2 confusing. In Section 5.1, why do the accuracy results for GPT-4 differ when comparing the left parts of Figure 3 and Figure 4?  I also found the experimental protocol in Section 5.2 somewhat perplexing. As I understand it, a constraint satisfaction problem described in natural language is first translated into a 3-SAT instance by a large language model (LLM) and then processed by a SAT solver. In this case, the “positive” results are not surprising at all, since the LLM is not being used to "reason" about the input problem; it is merely parsing the problem into a CNF expression.

**Contribution:** Based on the comments above, the main contribution of this paper is essentially limited to a statistical analysis of the accuracies of large language models (LLMs) on random instances of 3-SAT. By breaking down the distribution of instances into three typical regions, most LLMs struggle to solve instances in all these regions, even when the number of variables $n$ is small. Notably, we cannot conclusively state that GPT-4 behaves similarly to complete or local SAT algorithms for random 3-SAT; even for very easy instances ($\alpha \rightarrow 0$), its performance in finding a solution hovers around 70%. Therefore, a natural extension of this analysis would be to investigate the behavior of LLMs, particularly GPT-4, on simpler constraint satisfaction problems. This would help clarify their reasoning abilities. To this point, I would suggest examining random Krom instances (2-SAT), random Horn instances (HORN), and the intersection of these propositional classes (2-HORN-SAT).

**Minor comment:** In Section 2.2, nowadays, complete SAT solvers use CDCL instead of DPLL (in its basic form).

**Questions:**

See above comments.

---

> ### Author Response · Authors · 2024-11-17
> **Response to Reviewer pLV9**
>
> We sincerely thank the reviewer for their valuable feedback and for dedicating their time and effort to reviewing our paper. We appreciate that the reviewer acknowledges our novelty.
>
> ```Positioning: Although I am convinced that the paper is conveying something new, the related work section could benefit from further elaboration```
> Thank you for the feedback. We have now revised our Related Works (Section 3) to incorporate a detailed comparison. We also request the reviewer to refer to the Common Rebuttal where we summarize our work in the context of Related Works.
>
> ```why do the accuracy results for GPT-4 differ when comparing the left parts of Figure 3 and Figure 4?```
> As stated in the Figure captions, settings of Figure 3 and 4 are different -- SAT-CNF and SAT-Menu, respectively
>
> ```In Sec 5.2 (LLM-Modulo) the “positive” results are not surprising at all, since the LLM is not being used to "reason" about the input problem; it is merely parsing the problem into a CNF expression.```
> **You’re absolutely right**. However, the surprising insight is that by decomposing the problem in the most straightforward manner possible into a (easy) parsing task and a hard reasoning task, we achieve significantly better results compared to using an LLM only. **This suggests that effective reasoning with LLMs should involve decomposing tasks easy LLM problems and hard symbolic problems (when possible), rather than solely relying on scaling models with more training data and compute for natural language reasoning**. We have added this to our Conclusion.
>
> ```Contribution: Based on the comments above, the main contribution of this paper is essentially limited to a statistical analysis of the accuracies of large language models (LLMs) on random instances of 3-SAT. ... a natural extension of this analysis would be to investigate the behavior of LLMs, particularly GPT-4, on simpler constraint satisfaction problems.```
> We refer the reviewer to the Common rebuttal section, specifically -- 4. What do we NOT do? Nevertheless, we still ran the experiments with GPT-4 on 2-SAT and have added a discussion in Sec 6, Page 9.
>  Additionally, we point the reviewer to our revised Related Works section (Section 3) where we elaborate on how important our empirical analysis is in the context of similar theoretical claims showing that LLMs cannot reason.
>
> ```complete SAT solvers use CDCL instead of DPLL (in its basic form).```
> Thank you, we have now added a line in the revised version (L 173-174)

---

> > ### Author Response · Authors · 2024-11-20
> > **Following up on our Rebuttal to Reviewer pLV9**
> >
> > Dear Reviewer,
> >
> > As the end of the discussion phase draws near, please let us know if you have any further questions or clarifications that we could provide. We would be happy to comply. Thank you for your time and consideration.

---

> > > ### Comment · Reviewer_pLV9 · 2024-11-20
> > >
> > > I appreciate the authors' responses. And yes, I have one final comment: While I agree that using random 2-SAT may be too restrictive for evaluating the reasoning abilities of LLMs, I still believe that random Horn-SAT is a suitable candidate. As you know, Horn-SAT is P-complete, which implies it is P-hard. It is closely related to finite tree automata and reflects natural forms of human-like deduction abilities. Additionally, and perhaps most importantly, several subclasses of Horn-SAT exhibit non-trivial phase transitions regarding the probability of satisfiability (Demopoulos and Vardi, 2005; Moore et al., 2007).
> > >
> > > **References:**
> > >
> > > - Demopoulos, Demetrios D., and Moshe Y. Vardi. "The Phase Transition in the Random Horn-SAT Problem." In Allon Percus, Gabriel Istrate, and Cristopher Moore (Eds.), *Computational Complexity and Statistical Physics*, 2005.
> > >
> > > - Moore, Cristopher, Gabriel Istrate, Demetrios D. Demopoulos, and Moshe Y. Vardi. "A continuous-discontinuous second-order transition in the satisfiability of random Horn-SAT formulas." *Random Structures and Algorithms* 31(2): 173-185, 2007.

---

> ### Author Response · Authors · 2024-11-22
> **Additional Response to Reviewer pLV9**
>
> Thank you for engaging with us in the discussion. We completely agree that after having analyzed the 3-SAT characteristics of LLMs, SAT fragments like Horn-SAT present a natural continuation of our investigation into the reasoning abilities of these models. However, we believe there is more to studying SAT fragments than simply running straightforward experiments for the following reasons
>
> **Broader Perspective**: Among the six maximally tractable fragments identified (Schaefer 1978), two are trivially satisfiable, while the remaining four present interesting structural and algorithmic properties. These include 2-SAT, XOR-SAT, Horn-SAT, and negative Horn-SAT. **These fragments represent distinct avenues of exploration, each requiring comprehensive and focused work due to their unique structural and algorithmic properties. Undertaking such an expansive analysis would require significant additional effort, far exceeding the scope of this paper.**
>
> **Practical Issues**: As Istrate (2002) points out, "*in the critical region of Horn-SAT, the number of clauses is exponential in the number of variables*." This exponential growth presents practical challenges for analyzing random Horn formulas, especially when clause length is unrestricted.
>
> We hope the reviewer can share our perspective that a further investigation of tractable fragments, including Horn-SAT, falls outside the scope of this study. We believe we have established a comprehensive experimental protocol for 3-SAT. **Extended studies could then follow an experimental protocol of the same spirit as presented here for random 3-SAT.**
>
> ---
>
> Schaefer, T. J. (1978,). The complexity of satisfiability problems. In Proceedings of the tenth annual ACM symposium on Theory of computing (pp. 216-226).
>
> Istrate, Gabriel. (2002) "The phase transition in random Horn satisfiability and its algorithmic implications." Random Structures & Algorithms 20.4 (2002): 483-506.

---

> > ### Comment · Reviewer_pLV9 · 2024-11-22
> >
> > In my previous response, I mentioned "several subclasses" of Horn-SAT. Demopoulos and Vardi explored $j$-$k$ Horn formulas, which consist of clauses that are either of length $j$ or length $k$. In this case, where $j < k$ and $k$ is fixed, the length of the Horn formulas is in $O(n^k)$, with $n$ representing the number of variables. Additionally, Moore et al. have examined a generalization of these classes that also imposes a limit on the length of the clauses. As I noted earlier, some of these classes display interesting and nontrivial phase transitions.

---

> > > ### Author Response · Authors · 2024-11-24
> > > **Results for Horn-SAT**
> > >
> > > We are sincerely thankful for continuously engaging with us. We believe this has greatly contributed to the improvement of our paper.
> > >
> > > We ran some more experiments with 1-2-HornSAT and 1-3-HornSAT. In both, we observed a notable performance drop for GPT-4 around the satisfiability threshold -- the point where the probability of satisfiability transitions from >0.5 to <0.5 -- highlighted in **Figure 14**. This suggests that while **GPT-4 performs robustly on NL-complete problems like 2-SAT, its effectiveness diminishes for higher complexity classes such as P-complete (Horn-SAT) and NP-complete (3-SAT)**. These observations align with the findings of  Peng et al. (2024), Li et al. (2024)  which suggest that multi-layer transformers cannot solve problems such as Derivability, 2-SAT, Horn SAT, and Circuit Evaluation unless L=NL. However, with *T*-CoT steps (where *T* scales polynomially with sequence length), can compute any function solvable by a polynomial-sized circuit.
> > >
> > > We have revised our Discussion and Conclusion sections to include the same. Moreover, we have added our dataset statistics and generation process for both 2-SAT and Horn-SAT in Appendix A.

---

> > > > ### Author Response · Authors · 2024-11-27
> > > > **Following up on Horn-SAT results with Reviewer pLV9**
> > > >
> > > > Dear Reviewer,
> > > >
> > > > We wanted to follow up to see if the suggested additions to the manuscript (2-SAT, Horn-SAT) have addressed your concerns. If you feel that the paper has benefited from these improvements, we hope that this might be reflected in your assessment.
> > > >
> > > > We thank you again for engaging with us and providing useful feedback. We're happy to provide further clarification if required.

---

### Official Review · Reviewer_XxxU · 2024-11-01

**Soundness:** 3
**Presentation:** 3
**Contribution:** 2
**Rating:** 6
**Confidence:** 3

**Summary:**

This papers studies the question of whether LLMs can reason. Towards this question the authors propose a new method for studying reasoning capability, by evaluating the performance of LLMs in deciding satisfiability of 3-SAT instances.

This is motivated by a classic observation on the likelihood of random 3-SAT instances being satisfiable or unsatisfiable depending on the ratio alpha of clauses to variables in the instance. When alpha is low, the instance is almost surely satisfiable, and when the ratio is high, it is almost surely unsatisfiable. In the middle, there is a small range of this ratio where the satisfiability of random 3-SAT instances is hard to predict from statistical information.

The authors then compare this behaviour to the performance of various LLMs on random 3-SAT instances. As their key finding, they observe that the performance of GPT-4 is much better in the case of low or high alpha, suggesting that the LLM answers based on statistical/structural information rather than reasoning.

**Strengths:**

- The paper proposes an approach beyond the common toy-like example based evidence often found LLM reasoning research and provides well-founded analysis of reasoning in LLMs. Beyond the 3-SAT   based analysis in particular, the general approach has potential to crate a broader range of LLM reasoning benchmarks. As such the submission could be important to this emerging area of research.

- As a secondary result, the submission demonstrate that LLMs are very capable of translating 3-SAt instances in textual form into input to SAT solvers. Although there is a small gap in the paper here in that it is not clear what happens when the input in the SAT-Menu format is provided that is not restricted to clauses with at most 3 terms, i.e., when it would be a general SAT instance beyond 3-SAT.

- The presentation is clear, the claims and methods are easily followable.

**Weaknesses:**

- The paper claims that these results counter previous works that suggest reasoning ability in LLMs. However, the results of this submission suggest that LLMs are effectively unable to reason about an NP-hard problem. Previous positive results such as those by Kojima et al. (2022) are on inherently simpler problems. The paper lacks an appropriate discussion on this mismatch and the role NP-complete problems in the current discourse on LLM reasoning ability.

- As mentioned above, I find the approach itself promising. But the paper lacks any discussion of building on this approach to provide a more complete picture on LLM reasoning limits. For example, it would be a natural next step to demonstrate similar behaviour of LLMs also for problems in lower complexity classes. But there is no discussion along these lines nor an attempt to frame the approach in a general fashion.

- The set of tested models seems slightly outdated for this submission cycle (GPT 4 Turbo, Gemini 1.0, Llama 2).

**Questions:**

- Why are instance in the dataset annotated with the number of satisfying models rather than just SAT/UNSAT information?

- Have you considered extending this approach by any problems beyond 3-SAT? While I understand that the phase transition on random 3-SAT wrt. alpha is the key motivating factor for choosing 3-SAT, similar situations could maybe also arise in classic combinatorial problems over random graphs.

---

> ### Author Response · Authors · 2024-11-17
> **Response to Reviewer XxxU**
>
> We thank the reviewer for taking the time and effort to review our paper and give an overall positive assessment of our paper (*clear presentation, goes beyond toy-like examples, potential to create a broader range of LLM reasoning benchmarks, LLM as capable translators*). Your feedback was extremely useful in revising our paper.
>
> ```The paper claims that these results counter previous works that suggest reasoning ability in LLMs. However, the results of this submission suggest that LLMs are effectively unable to reason about an NP-hard problem. Previous positive results such as those by Kojima et al. (2022) are on inherently simpler problems...```
> We request the reviewer to refer to our Common Rebuttal where we summarize how existing benchmarks conflate reasoning with knowledge and potentially inflate performance due to data contamination. Moreover, we contend that previous results cannot be classified as inherently simpler problems without analyzing their alpha values (*precisely because these problems cannot be rigorously evaluated for their inherent hardness, without mapping them onto a formal representation...*).
>
> ``` I find the approach itself promising. But the paper lacks any discussion of building on this approach to provide a more complete picture on LLM reasoning limits. For example, it would be a natural next step to demonstrate similar behaviour of LLMs also for problems in lower complexity classes.```
> Once again, we refer the reviewer to the Common rebuttal section (4. What do we NOT do?). Nevertheless, we performed the suggested experiment and added this plot to the Appendix (Figure 13).
>
> ```The set of tested models seems slightly outdated for this submission cycle (GPT 4 Turbo, Gemini 1.0, Llama 2)```
> Indeed these are not from the latest generation of LLM. Due to cost constraints, we refrained from re-running our experiments on more expensive models. Nevertheless, **our experiments are comprehensive in that we compare multiple LLM implementations clearly showing that LLMs struggle with reasoning in terms of solving 3-SAT formulas** and **it is unlikely that results would change drastically even on the most recent LLMs**. We invite the broader research community, particularly those with more extensive resources at their disposal, to investigate these possibilities further.
>
> ```Why are instance in the dataset annotated with the number of satisfying models rather than just SAT/UNSAT information?```
> This is used for determining the satisfiability ratio as discussed in Figure 3 [Right].
>
> ```Have you considered extending this approach by any problems beyond 3-SAT? While I understand that the phase transition on random 3-SAT wrt. alpha is the key motivating factor for choosing 3-SAT, similar situations could maybe also arise in classic combinatorial problems over random graphs.```
> **That's an excellent point!** Certain combinatorial problems (like multiplication, dynamic programming etc.) were already explored in existing works like Dziri et al., 2023 where they show that the reasoning abilities of LLMs drop with an increase in problem size and depth. As stated in our revised Related Works section, we supplement this work by showing that it is the inherent hardness (and not problem depth/size) that determines performance -- and that -- problems with huge depths and sizes can also lie in easy regions.
> **We also emphasize that 3-SAT, as a prototypical NP-complete problem, serves as a representative testbed for reasoning**. Many other combinatorial problems, such as graph coloring, can be reduced to 3-SAT. Therefore, in theory, our findings should generalize to these related problems.
>
> [2] Dziri et al., Faith and fate: Limits of transformers on compositionality, NeurIPS 2023

---

> > ### Comment · Reviewer_XxxU · 2024-11-21
> >
> > I thank the authors for their thorough response.
> >
> > `[...] existing benchmarks conflate reasoning with knowledge and potentially inflate performance due to data contamination. Moreover, we contend that previous results cannot be classified as inherently simpler problems without analyzing their alpha values (precisely because these problems cannot be rigorously evaluated for their inherent hardness, without mapping them onto a formal representation...) `
> > While I agree with the point being made here I feel like it is somewhat past my original worry. The concern was that the statements in the paper were too broad, regarding the behaviour on 3-SAT as a refuation to other positive results on LLM reasoning. Of course some formal representation is necessary to formally study them, but it remains elusive to me why these problems are hard enough to necessitate an NP-hard formal representation.
> >
> > `[...] We also emphasize that 3-SAT, as a prototypical NP-complete problem, serves as a representative testbed for reasoning. [...]`
> > While 3-SAT has classically been *the* canonical NP-complete problem, we know from parameterized complexity that there are still significant differences in complexity between different NP-hard problems. Vertex-cover number, dominating set number, or even FPT fragments of 3-SAT could provide an interesting extension to the findings of the paper.
> >
> > But beyond NP-hardness, the thought (admittedly too implicit) behind my original question was to study the same for complete problems for other complexity classes. I appreciate the new results for 2-SAT in the new draft,  as another reviewer suggested Horn-SAT for P would be another interesting direction.

---

> ### Author Response · Authors · 2024-11-22
> **Additional Response to Reviewer XxxU**
>
> We thank the reviewer for engaging in the discussion.
>
> With 3-SAT, our primary goal is to *stress-test* LLMs in tackling harder combinatorial problems and evaluate their potential as substitutes for traditional solvers in planning and other combinatorial tasks -- **something that is increasingly gaining popularity**.  **As the reviewer rightly points out**, tractable fragments like **Horn-SAT, with linear complexity, could be used to extend our phase-transition study to comparatively easier problems**, although 3-SAT phase transitions are generally considered to be more prominent and interesting.
>
> In fact, there are six maximally tractable SAT fragments identified by Schaefer (1978). While two are trivially satisfiable, four present interesting structural and algorithmic properties. These include 2-SAT, XOR-SAT, Horn-SAT, and negative Horn-SAT. As we have pointed out in our response to Reviewer ```pLV9```, **analyzing each of the fragments comprehensively -- as we have done for 3-SAT -- merits expansive and focused effort due to their unique structural and algorithmic properties, far exceeding the scope of this paper.**
>
> Please also refer to our answer to Reviewer pLV9 where we also mention **practical issues that one might have to consider for Horn-SAT**. We hope the reviewer can share our perspective. We are grateful for the opportunity to engage further.
>
> ---
>
> Schaefer, T. J. (1978,). The complexity of satisfiability problems. In Proceedings of the tenth annual ACM symposium on Theory of computing (pp. 216-226).

---

> > ### Author Response · Authors · 2024-11-24
> > **Added results for lower complexity classes**
> >
> > We are sincerely thankful for the ongoing discussion that has helped improve our paper.
> >
> > In addition to our 2-SAT results, we have now added results on Horn-SAT. This is in response to the Reviewer's suggestion to perform similar analysis on lower-complexity classes.
> >
> > As shown in Figure 13 (2-SAT) and Figure 14 (1-2-HornSAT and 1-3-HornSAT), GPT-4 performs robustly on NL-complete problems like 2-SAT, its effectiveness diminishes for higher complexity classes such as P-complete (Horn-SAT) and NP-complete (3-SAT). These observations align with the findings of Peng et al. (2024), Li et al. (2024) which suggest that multi-layer transformers cannot solve problems such as Derivability, 2-SAT, Horn SAT, and Circuit Evaluation unless L=NL. However, with T-CoT steps (where T scales polynomially with sequence length), can compute any function solvable by a polynomial-sized circuit.
> >
> > For reference, please see our revised Discussion and Conclusion sections. Moreover, we have added our dataset statistics and generation process for both 2-SAT and Horn-SAT in Appendix A.

---

### Official Review · Reviewer_eHD7 · 2024-11-03

**Soundness:** 3
**Presentation:** 3
**Contribution:** 2
**Rating:** 5
**Confidence:** 4

**Summary:**

The authors study the reasoning capabilities of various LLMs by studying how well they can solve the Boolean satisfiability problem. The authors consider two different encodings of 3-SAT instances to be given as input to LLMs. An encoding to a natural language problem (they construct a problem related to a group of people ordering food items with given constraints) and by directly imputing formulae in 3-CNF to the LLMs.

They generate several datasets of CNFs by using different fixed parameters for the ratio of the number of clauses and variables. By varying this constant the authors can control the proportion of satisfiable formulae in their datasets.

They also consider the task of using LLMs to transform inputs in their format to a format understandable by a SAT solver.

In general, they find that LLMs cannot solve 3-SAT, but that they can transform inputs in their format to a format understandable by a SAT solver.

**Strengths:**

It is worthwhile to experimentally test the limitations of LLMs via problems arising from complexity theory. I liked the fact that the authors consider the phase transition related to 3-SAT. The discussion on related works is thorough.

**Weaknesses:**

The results are not surprising, as in general, current LLMs are not expressive enough (as mentioned by the authors) to decide 3-SAT. Moreover, LLMs have been recognised to be successful in transforming formats of diverse data, and hence it comes with no surprise that the integration of LLMs with SAT-solvers is successful. In summary, the authors only test how well various LLMs can solve 3-SAT. This kind of work is an excellent topic for a student project, but in my opinion does not suffice to be publishable in a top machine learning conference.

There is not much theoretical contribution in the submission. While it is natural to consider 3-SAT as a problem to solve and to use different encodings to LLMs, there is nothing theoretically novel there. The technical contribution is to generate these inputs (taking the phase transformation of 3-SAT in mind) and tabulate the results with respect to different LLMs. I do not think that these contributions suffice for a publication in a top general conference in machine learning.

**Questions:**

None.

---

> ### Author Response · Authors · 2024-11-17
> **Response to Reviewer eHD7**
>
> Indeed, as pointed out by the reviewer this is not a theoretical work but an empirical study of the reasoning capabilities of LLMs. However, our empirical findings complement existing theoretical results. Especially, as these theoretical results do only talk about worst-case complexity and do not provide more fine-grained statements.
>
> Furthermore, viewing our study in a historical context might also be helpful to understand the significance. Specifically, while it had been known that 3-SAT is NP-complete. It was still a surprising empirical finding that 3-SAT formulas undergo a phase transition and that this phase transition is correlated with the hardness of specific problem instances. Following the reviewer's argumentation this would also simply amount "to generate these inputs [...] and tabulate the results".
>
> The surprising finding of our study, we would argue, is that when solving random 3-SAT with LLMs we observe a dip in performance that correlates with the phase transition present in random 3-SAT instances. To us, this was a rather surprising finding even more so that we did not observe the same behavior for all LLMs. We do not believe that existing works, theoretical or empirical, have made this point.
>
> As for the reviewer's comment on the previous success of transforming data formats using LLMs, our experimental setup is not intended to show that this is possible but to show that decomposing the problems into a part that is easily solvable for an LLM and a part that can be solved by a symbolic solver results in an effortless improvement over solving the problem naively with an LLM. This gives a strong direction for future work that efforts of building models that can process natural language and that can reason should be directed towards such problem decomposition rather than hacking reasoning into LLMs. We again do not believe that this has been shown clearly in any previous study.
>
> Finally, we would like to make a general remark on the reviewer's opinion on experimental work. We find it unfortunate to see experimental work being held in such low esteem and being characterized as merely setting up an experiment and tabulating the measurements. This disrespects the hard work going into conceiving and setting up the experiment in the first place and performing the appropriate measurements. While pure empirical studies in computer science have traditionally not been a common technique to advance the field, we hold the opinion that with the advent of large artificial artifacts (e.g. LLMs), we ought to adopt techniques from the natural sciences. This trend can also be observed in a series of recent papers from the computer science community (Fan et al. 2023, Mirzadeh et al. 2024) and the physics community (Marino 2024). Notably, Mirzadeh et al., which was made public after the ICLR deadline, state in their abstract "We hypothesize that this decline is due to the fact that current LLMs are not capable of genuine logical reasoning; instead, they attempt to replicate the reasoning steps observed in their training data."
>
> Fan, L., Hua, W., Li, L., Ling, H., & Zhang, Y. (2023). Nphardeval: Dynamic benchmark on reasoning ability of large language models via complexity classes. arXiv preprint arXiv:2312.14890.
> Mirzadeh, I., Alizadeh, K., Shahrokhi, H., Tuzel, O., Bengio, S., & Farajtabar, M. (2024). Gsm-symbolic: Understanding the limitations of mathematical reasoning in large language models. arXiv preprint arXiv:2410.05229.
> Marino, R. (2024). Fast Analysis of the OpenAI O1-Preview Model in Solving Random K-SAT Problem: Does the LLM Solve the Problem Itself or Call an External SAT Solver?. arXiv preprint arXiv:2409.11232.

---

> > ### Comment · Reviewer_eHD7 · 2024-11-18
> > **Response to response**
> >
> > Thank you for your thorough response.
> >
> > Regarding your comment on the phase transition of 3SAT. The results there uncovered an interesting behaviour of a central problem in complexity theory. If no such behaviour had been discovered, then indeed that work would have been mostly unpublishable tabulation of data. Coming back to your results, in my opinion, you have not sufficiently argued that the data you have obtained reveals something new and fundamental regarding the capabilities of LLMs.
> >
> > If I interpreted the confusion matrices of Figure 11 (revised version) correctly, GPT-4 is the only LLM that slightly beats a coin flip, while all the other LLMs are highly skewed to output "SAT" independent of what the input is. Perhaps a more detailed analysis here on whether the value of alpha has an effect on the confusion matrices would have revealed something interesting.
> >
> > Overall, I think one big issue is that, since all the LLMs tested are so bad at solving 3SAT, it is quite challenging to obtain interesting and meaningful results from the data, unfortunately.
> >
> > I do not wish to belittle experimental research in computer science, and I do recognise that creating data and running experiments is hard work. Nevertheless, in this case and in my opinion, the results drawn from the experiments are not strong enough for publication in a top general conference in machine learning.

---

> > > ### Author Response · Authors · 2024-11-18
> > > **Requesting Reviewer to provide specific references**
> > >
> > > ```The results are not surprising, as in general, current LLMs are not expressive enough (as mentioned by the authors) to decide 3-SAT. ```
> > > We would like to thank the reviewer for engaging in the discussion and hope to respond to their criticism. However, without concrete references to prior works that supposedly have already demonstrated our results, we're unsure how to respond to the reviewer's comments adequately. **If these results (i.e. *LLMs struggle to perform formal reasoning for inherently hard problems that require search*) were indeed obvious, we question why benchmarks for LLM reasoning continue to be a standard measure for evaluating their reasoning capabilities.** We would also like to point out that our paper includes detailed comparisons showing how our work complements the theoretical literature on LLM reasoning. We kindly request the reviewer to support their claim or refute our contributions with concrete references. Please note that our core findings have been acknowledged by reviewers ```oKUj, XxxU, and pLV9```.
> > >
> > > We will now do our best to interpret the reviewer's comments. Regarding the statement: ```In my opinion, you have not sufficiently argued that the data you have obtained reveals something new and fundamental regarding the capabilities of LLMs,``` we wish to reiterate our experimental findings, as they appear to have been overlooked:
> > >
> > > 1) Our experiments clearly demonstrate that transformer architectures can exhibit phase-transition-like behavior when solving reasoning problems. We observed this with GPT-4, and to the best of our knowledge, this behavior has not been previously documented in the literature. (Sec 5)
> > > 2) We made an important observation about the satisfiability ratio of random 3-SAT formulas: formulas with more satisfying assignments tend to be easier for LLMs to solve. This holds true for both the easy and hard regions. Again, this has not been previously shown in the literature. (L 348-354)
> > > 3) While prior works have shown that LLM performance for logical reasoning degrades as the problem size increases (e.g., NPHard Eval and Dziri et al., 2023), our study refines this understanding by identifying that this decline is linked to the inherent difficulty of reasoning in the hard region, independent of problem size or depth. Please refer to our revised Related Works section.
> > > 4) We would also like to reiterate our finding that performing the simplest problem decomposition possible into an LLM-friendly part and a solver-friendly part considerably boosts performance. Thereby motivating avenues for future work on reasoning with LLMs.
> > >
> > >
> > > ```If I interpreted the confusion matrices of Figure 11 (revised version) correctly, GPT-4 is the only LLM that slightly beats a coin flip, while all the other LLMs are highly skewed to output "SAT" independent of what the input is.```
> > >
> > > We believe the reviewer might be dismissing our study too readily by implying that the performance on SAT-Decision is merely noise, with LLMs mostly predicting 'SAT' regardless of the input. However, we conducted a separate study on SAT-Search, in which the LLM generates solutions to satisfiable problems. We observed a similar phase transition behavior in SAT-Search. Furthermore, our findings from SAT-Search are supported by our satisfiability ratio study (L 348-354), showing that LLMs are more likely to find a solution when the number of satisfying assignments is high, and vice versa.
> > >
> > > ```one big issue is that, since all the LLMs tested are so bad at solving 3SAT, it is quite challenging to obtain interesting and meaningful results from the data```
> > > To us, this is again a surprising statement as the tests LLMs and especially GPT4 perform well outside of the hard region. Nuancing this a bit we also see a qualitatively different behavior between different transformer-based LLMs (GPT4 vs. the others). This is again an observation that has not been made in any prior work.

---

> > > > ### Comment · Reviewer_eHD7 · 2024-11-25
> > > > **Response to the authors**
> > > >
> > > > Thank you for your detailed answers. I appreciate your thorough discussion on related works (and the additions made due to comments of other reviewers).
> > > >
> > > > I agree with Reviewer oKUj in that your position on whether you think your results support or not that LLMs have emergent deductive abilities remains quite unclear (this is not really important, but somewhat interesting).
> > > >
> > > > I think some more analysis could be made to isolate the increased benefit of using LLMs to flipping a suitably balanced dice. From Figure 4, one can read that for the hard region GPT-4 is as good picking a satisfying assignment than a random pick, while in the easy region there is a measurable difference. After discussions and reading the revised version, as well as other discussions here, I have re-evaluated my stance that your data reveals something interesting on the current capabilities on LLMs. Hence, I will increase my evaluation accordingly. Though, I still have concerns on the impact and importance of the study.
> > > >
> > > > Minor comment: You write in Figure 9 that "both variants fall under the same complexity class, ... the decision problem than the search problem". This is not really true. The decision problem is NP-complete, while the search problem is characterised by a function complexity class FNP.

---

> > > > > ### Author Response · Authors · 2024-11-27
> > > > > **Thank you Reviewer eHD7**
> > > > >
> > > > > We sincerely thank the Reviewer for raising their scores and are glad that the discussion and new results have helped further clarify the significance of our findings. Building on the Discussion in Section 6 of our paper, we would like to reiterate and clarify the following points:
> > > > > ```
> > > > > ... GPT-4’s apparent reasoning capabilities (in the easy regions) is due to the presence of statistical features that it can leach onto. For instance, GPT-4 may oversimplify and deem a problem unsatisfiable due to the high number of input tokens, which often works for overconstrained formulas (see Appendix B). Conversely, in the hard region, the drop in performance can be attributed to GPT-4’s – and by extension current transformer-based LLMs’ – inability to reason according to Bottou’s definition.
> > > > > ```
> > > > > Specifically, we argue that reasoning in the hard region aligns with Bottou's (and, by extension, our) definition of reasoning, which involves the algebraic manipulation of knowledge. Since GPT-4 and similar LLMs fail to demonstrate reasoning in the hard region, we conclude that they cannot truly reason. While we state this up front in our abstract (L 18-19), we will ensure this stance is articulated more strongly in our revised discussion.
> > > > >
> > > > > Additionally, based on our new results on 2-SAT and Horn-SAT, we state the following:
> > > > > ```
> > > > > ...This suggests that while GPT-4 performs robustly on NL-complete problems like 2-SAT, its effectiveness diminishes for higher complexity classes such as P-complete (HornSAT) and NP-complete (3-SAT) ...
> > > > > ```
> > > > >
> > > > > Thank you once again for your valuable comments and suggestions. We will ensure these points are incorporated into our manuscript.

---

### Official Review · Reviewer_oKUj · 2024-11-03

**Soundness:** 3
**Presentation:** 2
**Contribution:** 3
**Rating:** 5
**Confidence:** 4

**Summary:**

The paper analyzes the algorithmic reasoning abilities of LLMs via the 3-SAT problem. The authors examine performance on instances of varying hardness as characterized by the phase transition of random 3-SAT. They test the LLM on three kinds of prompt: a representation of an integer CNF formulation, a natural language translation of the same, and a prompt that just asks the LLM to translate a natural language instance into LaTeX. They find that LLMs fail to robustly solve these problems, and SoTA systems perform worse on harder problems, but that--across many models--performance is much lower than the phase transition analysis would imply in high alpha regions.

**Strengths:**

1. Contrasting previous work, the authors clearly define which notion of "reasoning" they are examining and choose a canonical, well-studied classical problem to examine LLM performance on.
1. The introduction does a good job of covering most of major, relevant LLM reasoning-related previous work, situating the current work in the landscape of both positive and negative results.
1. The authors construct their evaluation set using a strong, well-studied random sampling procedure for 3-SAT problems, ensuring the distributional validity of their results, and allowing for more detailed analysis grounded in previous work.
1. The authors analyze much more than just accuracy, considering not only performance around the critical region, but also relative to another proxy for difficulty: the satisfiability ratio, thus strengthening their results.
1. The authors demonstrate how to boost performance in some cases by combining the LLM with an external solver.
1. The authors do their tests across a number of different models.

**Weaknesses:**

## Issues with Natural Language Prompt formulation

If I'm interpreting the SAT-MENU prompt correctly, then I believe it violates commonsense. The prompt asks for a list of orderable and a list of non-orderable foods that satisfies a group of people, or equivalently an assignment of "orderable" or "not orderable" to each food in a pre-specified list of foods. The immediate assumption is that satisfying this group of people has something to do with the actual act of ordering foods for the group somewhere down the line, e.g. that these lists will be used for ordering later on. However, given this or a similar assumption, the problem specification leads to some rather absurd possibilities.

Consider "Jay dislikes pizza, pie, and nachos. Mary likes pizza and pie, but dislikes nachos." Is there an orderable list and non-orderable list which satisfies both Jay and Mary, using only pizza, pie, and nachos? A commonsensical answer would be no, because Jay doesn't like anything on the menu, but the structure of the problem is such that Jay is happy as long as one of the things he doesn't like isn't ordered. In particular, we have the following absurd-looking satisfying assignment: "orderable: nachos, pie. non-orderable: pizza" in which Jay is happy because pizza isn't orderable; whereas nachos, disfavored by both, *are* orderable. I know this is somewhat subjective, but this makes SAT-MENU more of a syntactic sugaring of the CNF prompt rather than a natural example of somewhere where this kind of reasoning needs to be done.

This likely speaks to a broader issue with the representations used here. The CNF formulation is a modeling convenience rather than a natural framing of real-world problems (see e.g. Stuckey 2013 "There Are No CNF Problems"). Analyzing SAT-CNF makes sense if we are interested in how well LLMs have acquired the ability to solve classical 3-SAT problems. SAT-MENU is the same problem with added distractors. Neither of them are well-justified as proxies for the kinds of natural language constraint satisfaction queries we might expect LLMs to actually be asked to solve. I would appreciate if the authors would clarify if the goal is to examine average-case reasoning performance (as alluded to in line 444) or to demonstrate the existence of a domain on which LLMs clearly fail to reason and resort to statistical shortcuts. If it is the former, I would be interested in seeing more plausible prompt reformulations.

## Complexity Class Claims
At lines 90-91 and 151, the authors claim that "logical reasoning, planning, and constraint satisfaction" can be reduced to 3-SAT. This is only true for limited forms of logical reasoning, e.g. the decision problem for first-order logic is in fact undecidable. Furthermore, planning also cannot in general be reduced this way: just the problem of plan existence (in STRIPS planning) is already PSPACE-complete. Note however that the (easier) *scheduling* phase is generally reducible to constraint satisfaction.

## Unclear Relationship to Cited Paper: Kambhampati 2024a
Lines 138-140 do not seem to match section 5.2. Emphasis mine:

> Additionally, we demonstrate how integrating LLMs with external **verifiers**, such as in the LLM-Modulo Frameworks (Kambhampati et al., 2024a), can enhance reasoning capabilities and improve performance on 3-SAT problems

Line 350 restates the claim:

> "The main idea is to augment LLMs with **critics and verifiers** (Lightman et al., 2024; Hazra et al., 2024b), recognizing the ability of LLMs as approximate idea-generators for problems as against directly solving them"

These quotes accurately reflect the LLM-Modulo framework as described in Kambhampati 2024a (quote from p6 of that paper):

> "LLM-Modulo architecture is a 'Generate-Test' one that involves LLMs interacting with the external critics/verifiers **rather than a LLMs being just frontends to external solvers**"

However, lines 355-356 describe the setup the authors tried in this paper, which contradicts both their previous summaries of their own work as well as the main idea of the framework they claim to be implementing. Section 5.2 describes how to use the LLM as a syntactic translator from the SAT-MENU format into one that MiniSAT can process, rather than using the LLM as a generator for proposed answers that are then filtered through sound verifiers/critics. Because of the problems presented (which are essentially already in CNF form) and the SAT solver in the loop, this seems to be a noisier analysis of the ability for the LLM to do syntactic translation, and doesn't tell anything about reasoning. (I say noisier because there is a chance that the generated translation was incorrect, but happened to have a satisfying assignment that also satisfies the real problem.)

If the authors are interested in the LLM-Modulo framework, perhaps one approach would be to compare how many generate-test iterations it takes for the model to output a satisfying assignment relative to the alpha region.

## Other Citation and Unclear Claim Issues/Nitpicks
1. The paper states that "emergent abilities have been shown to emerge" in line 39, but later (line 83) cites a paper claiming that emergent abilities are "a mere mirage," making the authors' position unclear.
1. Lines 80-81 cite Dziri 2023 for a claim about architectural limits of transformer layers, but that paper is an empirical evaluation of pre-trained models together with some (very broad) theoretical claims that are applicable to any autoregressive model, not just transformers. This citation doesn't seem to be relevant here and should likely be removed.
1. Line 182-183: Olausson 2024 and Liu 2023 are cited in the context of combining LLMs with verifiers and heuristics, but the systems proposed in those papers (LINC and LLM+P) combine translator LLMs with *solvers*, not verifiers. It's unclear why Lightman 2024 is cited, as it seems to be about process supervision at train time, rather than combining an LLM with a verifier at inference time.
1. Line 340: in-text citation was incorrectly formatted (citet should be citep).
1. The final paragraph of the conclusion (line 468-471) makes claims that seem unrelated to and unexamined by the content of the paper's body.

**Questions:**

1. I notice that, in figure 3, GPT-4 seemingly significantly worse than random guessing on the SAT decision problem around the critical point (which would imply for instance, that taking the opposite of its answer would be a better algorithm for the decision problem, giving about 70% accuracy right at alpha). Is this effect consistent/significant/predictable? Have the authors looked at why this is?
1. Given that these prompts are CoT prompts, have the authors looked at what sort of procedure the LLM claims to be following (and if it matches with the given examples)? Specifically, I'm curious if we can distinguish the LLM results from what we would get if instead we tested a noisy reasoner--a very simple example would be an implementation of DPLL where every each search step has some fixed epsilon probability of failure.

---

> ### Author Response · Authors · 2024-11-17
> **Response to Reviewer oKUj**
>
> We thank the reviewer for their detailed comments and analysis which greatly helped improve the narrative of the paper. We appreciate the time and effort put into this review and look forward to engaging further if necessary.
>
> ```SAT Menu task violates commonsense ... Neither of them are well-justified as proxies for the kinds of natural language constraint satisfaction queries we might expect LLMs to actually be asked to solve.```
> **We respectfully disagree with the premise**, that LLMs would be expected to only solve tasks that adhere to commonsense norms. In fact, tasks in the real-world (like travel planning, and robotic task planning) often involve a combination of commonsense reasoning (that requires knowledge retrieval) and logical or deductive reasoning (that requires algebraic manipulation and composition of the knowledge based on Bottou's definition). Our study, as outlined in the Introduction, focuses explicitly on evaluating the latter—logical and deductive reasoning—without conflating it with commonsense reasoning, which is a common limitation in many existing benchmarks. To this end, we believe this design choice is not a bug but a feature, allowing us to measure -- in a more controlled setting -- the extent to which LLMs rely on statistical patterns versus actual reasoning. **Our setting highlights the challenges of reasoning in isolation from context-dependent or knowledge-based shortcuts**, as long as our prompts outline a clear objective and have all the required information to solve the task.
>
> **Notably, the overall characteristics remain consistent across different prompts (SAT-Menu, SAT-CNF), despite the LLMs employing observably different reasoning strategies.** For instance, with CNF inputs, the LLM often mimics DPLL-like behavior involving backtracking and occasionally attempts local search. In contrast, with natural language menu inputs, the LLM generally struggles to interpret the underlying CNF formula and resorts to trial-and-error reasoning to find a solution.
>
> Indeed, one of the goals of the paper is to complement theoretical findings on the expressive power of LLMs by using a more fine-grained empirical analysis. Following our argumentations above on commonsense we believe that our experimental evaluation achieves this.
>
> ```... the authors claim that "logical reasoning, planning, and constraint satisfaction" can be reduced to 3-SAT. This is only true for limited forms of logical reasoning```
> Indeed, this is a slight imprecision on our part as we were implicitly referring to propositional reasoning  (e.g. optimal policy for MDPs is NP-complete, and it fits into the planning definition). We have clarified this in the paper. Thank you.
>
> ```Unclear Relationship to Cited Paper: Kambhampati 2024a```
> We have revised this section to improve clarity -- once again, thank you for pointing this out. We acknowledge the noise introduced when using an LLM + solver approach. However, our experiment is not designed to improve or comment upon the reasoning capabilities of LLMs. What it does, however, is show that decomposing the problem at hand into an LLM-friendly problem and a solver-friendly problem in the simplest way possible drastically improves performance. This gives a strong suggestion for building future reasoning agents. Namely, equipping LLMs with external solvers instead of performing reasoning in the LLM itself. See also our general comment.
>
> ```GPT-4 seemingly significantly worse than random guessing on the SAT decision problem around the critical point ... that taking the opposite of its answer would be a better algorithm for the decision problem```
> We would like to reiterate that the goal of our study is not to solve 3-SAT (as stated in Related Works), but rather to establish the reasoning abilities of LLMs. Furthermore, flipping the answer for certain regions amounts to knowing the alpha value to flip at. The problem here is that this alpha value varies from LLM to LLM. Furthermore, in practical reasoning settings, this alpha value is not known. We only have access to it because we assess the reasoning capabilities on random 3-SAT.
>
> **As for why this happens**: A plausible explanation is that the hard region demands a deeper search and more reasoning steps. Analysis of GPT-4's outputs reveals that the model often takes a "lazy" approach, **either giving up and suggesting delegating the problem to a solver** (Box 5) or **providing only a rough outline of the solution** for the user. This was observed for both SAN-CNF as well as SAT-Menu where it concludes (*... Considering the complex preferences, a comprehensive computational approach is warranted here because manual trial and error would likely be extremely time-consuming and prone to error. In the absence of a computational tool to analyze this vast dataset and given the mutually exclusive preferences, we will assume that no such satisfactory combination exists ...*). We have added more such examples to Appendix B.

---

> > ### Author Response · Authors · 2024-11-17
> > **Response to Reviewer oKUj (Continued)**
> >
> > ```Given that these prompts are CoT prompts, have the authors looked at what sort of procedure the LLM claims to be following (and if it matches with the given examples)? ```
> >
> > We observed the following behaviors in the generated outputs, including chain-of-thought (CoT) reasoning:
> >
> > **Diverse Reasoning Techniques**: GPT-4 employs varying reasoning techniques depending on the prompt type (SAT-CNF vs. SAT-Menu) and even adapts its approach across individual problems within the same prompt type.
> >
> > **SAT-CNF Reasoning**: The dominant strategy involves backtracking, as illustrated in Box 5. Occasionally, GPT-4 employs local search, where it assigns items to "orderable" and "not-orderable" lists and iteratively modifies these based on detected conflicts (e.g., *... We can create two sets for liked and disliked items and then compare them to find any conflicts. Let's begin by creating a list of all the likes and dislikes to identify conflicts...*).
> >
> > **SAT-Menu Reasoning**: The primary strategy here is trial-and-error. Occasionally, GPT-4 applies heuristics such as the Maximum Occurrence in Minimum-sized clauses (MOM) heuristic to prioritize variables appearing most frequently in the smallest clauses (e.g., *...We start by making a tally of how many people like or dislike each food item... If we put 'macaron' on the 'orderable' list, we will satisfy many people who like it...*).
> >
> > **"Lazy" Solutions**: As previously noted, GPT-4 often produces "lazy" solutions in many cases, either providing an outline of how to solve the problem or asking to be delegated to a solver.
> >
> > We have added this to Appendix B of our revised version.

---

> > > ### Author Response · Authors · 2024-11-20
> > > **Following up on our rebuttal**
> > >
> > > We thank the reviewer for their detailed feedback and their overall positive assessment of our contributions.
> > >
> > > Since the end of the discussion phase is drawing near, we were wondering if there are any further questions or clarifications you'd like us to address. We'd gladly provide additional details if needed.

---

> ### Author Response · Authors · 2024-11-27
> **Following up with Reviewer oKUj**
>
> Dear Reviewer,
>
> Since we didn't receive any further questions from you, we wanted to follow up again to see if our responses to your queries and the corresponding additions to the manuscript have addressed your concerns. We have also added results on 2-SAT and Horn-SAT as suggested by Reviewers ```XxxU, pLV9```. If you feel that the paper has benefited from these improvements, we hope that this might be reflected in your assessment.
>
> That said, we'd be more than happy to engage further or provide any additional clarifications.
>
> We deeply appreciate your time and effort to review our work.

---

### Author Response · Authors · 2024-11-17
**Common Rebuttal by Authors**

We sincerely thank all the reviewers for their valuable feedback, which has greatly helped us improve our work. We are elated that the reviewers acknowledge our core contributions (```oKUj,pLV9```) and their contextualization within existing works (```oKUj```), and find our presentation clear (```XxxU```) and experiments diverse (```oKUj,XxxU```).

To streamline the review process, we recap the key points from our paper:

**1. What are the problems with existing reasoning benchmarks?**
**Current benchmarks often conflate commonsense reasoning** (which involves knowledge retrieval), **with logic and deductive reasoning** (which requires algebraic manipulation of knowledge as per Bottou’s definition). This conflation makes it challenging to isolate the logical reasoning abilities of LLMs. Moreover, recent findings (e.g., Zhang et al., 2024) highlight issues such as dataset contamination inflating performance metrics. Logical reasoning is critical for real-world applications like travel planning and robotic task execution, where isolated evaluation of reasoning without relying on context-dependent or knowledge-based shortcuts is essential. We argue that previous works do not conclusively deal with inherently simpler problems, precisely because these problems cannot be rigorously evaluated for their inherent hardness, without mapping them onto a formal representation (like 3-SAT). **We have added more clarity regarding this distinction in L90-98.**

**2. How do we overcome this problem?**
We start by **defining reasoning in terms of the 3-SAT problem** -- a prototypical NP-complete problem -- specifically **analyzing phase transitions in 3-SAT**. These phase transitions are well-established indicators of *inherent problem hardness*. Easy regions: Problems in these regions are solvable using statistical features, often without requiring explicit search. Hard regions: In these regions, no known heuristics exist, and statistical shortcuts fail. Solving problems here necessitates explicit search, as LLMs cannot rely solely on pre-trained knowledge or statistical patterns. We observed that LLMs generally struggle in the hard region which indicates that they fail to perform search (Figure 3). Conversely, their relatively better performance in easy regions suggests reliance on statistical features and reasoning shortcuts rather than genuine deductive reasoning. We also show how such reasoning tasks can be solved using a straightforward integration of LLM + Solver (Sec 5.2). This suggests that **effective reasoning with LLMs should involve decomposing tasks when possible, rather than solely relying on scaling models with more training data and compute for natural language reasoning.**

**3. How is our work aligned with recent papers showing the reasoning limitations of LLMs?**
Our work complements recent studies on the reasoning limitations of LLM
* Dziri et al. (2023): Performance declines with increasing task complexity (size, depth). We extend this, attributing declines to inherent problem hardness, not merely size or depth.
* Li et al. (2024): Demonstrated how T-CoT steps can extend transformer reasoning abilities up to problems solvable by Boolean circuits of size T. Our findings on GPT-4 reveal higher token generation in hard regions, suggesting apparent reasoning despite poor performance.
* Merrill & Sabharwal (2023), Peng et al., 2024: Theoretical performance bounds for multi-layered transformer architectures focus on worst-case scenarios but offer limited insights into average-case complexities. We bridge this gap empirically. **We elaborate on this in our revised Related Works section.**

**4. What do we NOT do?**
Our goal is not to design a 3-SAT solver using LLMs but to assess their reasoning abilities. Why 3-SAT? It is unclear how a similar empirical analysis could be performed for lower complexity classes as well. While phase transitions are also exhibited by random 2-SAT, which can be solved in polytime, it is barely detectable (Goerdt 1996). We observed the same with GPT-4 and have added this plot as Figure 13. **3-SAT is really the prototypical problem for NP-completeness which shows pronounced phase transition characteristics.**

---

[1] Zhang et al., A careful examination of large language model performance on grade school arithmetic., 2024
[2] Dziri et al., Faith and fate: Limits of transformers on compositionality, NeurIPS 2023
[3] Li et al., Chain of thought empowers transformers to solve inherently serial problems, ICLR 2024
[4] Merrill & Sabharwal, The parallelism tradeoff: Limitations of log-precision transformers, TACL, 2023
[5] Peng et al., On limitations of the transformer architecture, 2024
[6] Goerdt, A threshold for unsatisfiability, 1996

---

> ### Author Response · Authors · 2024-11-17
> **List of revision included in the paper**
>
> **Besides revising parts of our paper, we made the following additions based on the reviews**:
>
> * Added 2-SAT search plot for GPT-4 (Appendix Figure 13) and a discussion of the same in Section 6.
> * Added a comparison for the number of generated tokens vs alpha, which is an empirical exploration of Li et al., 2024. The findings reveal that GPT-4 generated tokens increase in the hard region (Experiments L370-375, Figure 12).
> * Added an analysis of generated output from GPT-4, including reasoning strategies used and interesting failure cases. (Appendix B)

---

### Meta-Review · Area_Chair_jS1q · 2024-12-23

**Metareview:**

**Summary:**
This paper studies reasoning ability of LLMs. For this purpose this paper experiments performance of LLMs in solving 3-SAT problems. Two different modes of presenting problem instances to LLMs are considered: one is SAT-CNF, where the prompts to be presented to an LLM are something like that shown in Box 3, and the other is SAT-Menu, where a SAT problem instance is reframed as a natural-language menu-selection problem as in Box 2. Two tasks are considered: in "SAT Decision" an LLM is asked to respond with "Yes/No" according to whether the presented instance is satisfiable or not, whereas in "SAT Search" an LLM should also return an assignment if the presented problem instance is SAT. The experimental results are compared with the known satisfiability threshold of random 3-SAT, which is $\alpha_c\approx4.27$.

**Strengths:**
The authors clearly define the notion of "reasoning" they examine, and take 3-SAT as a canonical, well-established problem from complexity theory, on which performance of LLMs is examined.

**Weaknesses:**
- Some reviewers expressed concern on the importance of the results.
- In addition to the concerns raised by the reviewers, I would add one more point. It is known, besides the satisfiability transition point $\alpha_c\approx4.27$, that there exist some other types of phase transition for 3-SAT as well (see, e.g., Krzakala et al. (2007)): for example, the dynamic phase transition point $\alpha_d\approx3.86$, above which the set of solutions decomposes into many disconnected clusters, and the condensation phase transition point (which is known to coincide with $\alpha_d$ for 3-SAT, and to be larger than $\alpha_d$ and smaller than $\alpha_c$ for $K$-SAT with $K\ge4$), above which the cluster sizes become uneven. One would expect that heuristic methods would be effective for $\alpha$ less than $\alpha_d$ but not for $\alpha$ larger than $\alpha_d$, making $\alpha_d$ a sound choice as the easy-hard boundary. (Note that these notions are defined in the asymptotic $n\to\infty$.) The authors should take into account these results in their analysis.

Krzakala et al., "Gibbs states and the set of solutions of random constraint satisfaction problems," PNAS, volume 104, pages 10318-10323, 2007.

**Reasons:**
Three reviewers rated this paper just below the acceptance threshold. My own evaluation is aligned with these reviewers in that, although this paper presents something interesting on reasoning abilities of LLMs, it would benefit from further revisions taking the review comments into account.

Additional points:
- Figure 2: Those plots (probability of satisfiability and solver running time versus $\alpha$) should depend on the number $n$ of variables, whose value is however not mentioned. Whether the probability of satisfiability is 1 or 0, which determines the hard region, should also depend on $n$ and the number of instances examined (CNF formulas): if more instances are examined one would have a narrower hardness region. Furthermore, the *true* probability of a random formula to be satisfiable is never equal to 0, since, given an arbitrary assignment, there always exists a formula which satisfies the given assignment, making the probability strictly positive for any $\alpha$.  It would make the use of the probability of satisfiability in defining the hardness region inappropriate.
- Figure 3 [Right]: Again, there should be $n$-dependence as above, which is not stated explicitly. I did not understand why the authors show two plots, one for "easy" and the other for "hard".
  - These two are not distinguished in Figure 7, which shows similar plots for other LLMs. It would make the significance of distinguishing "easy" and "hard" in Figure 3 unclear.
  - I did not understand either why the ranges of satisfiability ratios for the two plots overlap. One would expect, as a general trend, that the satisfiability ratio monotonically decreases as $\alpha$ becomes larger, so that the "easy" plot should have a gap in the satisfiability ratio values that corresponds to the range of the "hard" region.
- Figure 6 right: I guess that the multi-peak structure of the distributions of $\alpha$ values shown here would be an artefact arising from use of a kernel density estimator. This figure would provide a wrong impression as if there are probability density functions for the values of $\alpha$.

**Additional Comments On Reviewer Discussion:**

Although all the reviewers acknowledged the approach adopted in this paper, some reviewers expressed their concern on the impact and the importance even after the author rebuttal and discussion between the authors and the reviewers.

---

### Decision · Program_Chairs · 2025-01-22

Reject